# Creb5 coordinates synovial joint formation with the genesis of articular cartilage

Cheng-Hai Zhang [1] ✉, Yao Gao [1], Han-Hwa Hung[2], Zhu Zhuo[3], Alan J. Grodzinsky[2] & Andrew B. Lassar [1] ✉

While prior work has established that articular cartilage arises from Prg4-expressing perichondrial cells, it is not clear how this process is specifically restricted to the perichondrium of synovial joints. We document that the transcription factor Creb5 is necessary to initiate the expression of signaling molecules that both direct the formation of synovial joints and guide perichondrial tissue to form articular cartilage instead of bone. Creb5 promotes the generation of articular chondrocytes from perichondrial precursors in part by inducing expression of signaling molecules that block a Wnt5a autoregulatory loop in the perichondrium. Postnatal deletion of Creb5 in the articular cartilage leads to loss of both flat superficial zone articular chondrocytes coupled with a loss of both Prg4 and Wif1 expression in the articular cartilage; and a non-cell autonomous up-regulation of Ctgf. Our findings indicate that Creb5 promotes joint formation and the subsequent development of articular chondrocytes by driving the expression of signaling molecules that both specify the joint interzone and simultaneously inhibit a Wnt5a positive-feedback loop in the perichondrium.

Vertebrate bones develop via endochondral ossification in which chondrocytes are fashioned into a transient cartilage template, which undergoes a highly regulated maturation process in growth plates to give rise to trabecular bone[1,2]. Notably, the perichondrium that surrounds the transient cartilage templates gives rise to either articular chondrocytes in the joint regions[3–5] or cortical bone in the shaft of the developing long bones[6,7]. It is not clear what transcriptional regulators control the formation of these distinct perichondrial cell fates. In striking contrast to chondrocytes in the cartilage core which undergo endochondral ossification, articular chondrocytes maintain an immature chondrogenic differentiation program; and give rise to permanent articular cartilage in synovial joints[8,9]. In addition to its relative permanence, articular cartilage can be distinguished from transient growth plate cartilage by expression of lubricin (a.k.a. Proteoglycan 4)[5], which is encoded by the *Prg4* gene. Prg4 is initially expressed in the articular perichondrium, which encases the ends of the developing long bones, and is subsequently expressed in the superficial-most layer of mature articular cartilage, but not by deeper layers of this tissue[10–15]. Fate mapping studies have established that Prg4-expressing cells in embryonic and early postnatal joints constitute a progenitor pool for all regions of the articular cartilage in the adult[3–5]. In contrast, cortical bone progenitors arise from more centrally located perichondrium[6,7]; and chondrocytes that undergo endochondral ossification, in the growth plates, arise from the cartilage core[16,17]. We have recently identified Creb5 as a transcription factor that is specifically expressed in bovine superficial zone articular cartilage and is required to promote Prg4 expression in response to TGF-β and EGFR signaling[18]. In this work we demonstrate that Creb5 plays an instrumental role in the formation of synovial joints, and in addition both directs the genesis of the articular cartilage and regulates the morphogenesis of the underlying epiphyseal ossification center by blocking a Wnt5a positive-feedback loop in the perichondrium.

[1]Department of Biological Chemistry and Molecular Pharmacology, Blavatnik Institute at Harvard Medical School, 240 Longwood Ave., Boston, MA 02115, USA. [2]Department of Biological Engineering, Massachusetts Institute of Technology, Cambridge, MA 02139, USA. [3]Bioinformatics Core, Department of Biostatistics, Harvard T.H. Chan School of Public Health, Boston, MA 02115, USA. ✉e-mail: Chenghai_Zhang@hms.harvard.edu; Andrew_Lassar@hms.harvard.edu

## Results

### *Creb5* drives gene expression in the joint interzone

In Po mice, expression of Creb5 RNA and protein is restricted to Prg4+ cells in the nascent joint; and is specifically expressed in the articular perichondrium but not expressed in the metaphyseal/diaphyseal perichondrium[18]. To further elucidate the role of Creb5 in formation of synovial joints, we assayed the expression of this gene during the early phases of synovial joint formation. RNAscope fluorescent in situ hybridization (FISH) in E13.5 mice embryos indicated that Creb5 is expressed in a broad domain of the developing forelimb autopod, and that its expression in the nascent shoulder and elbow joints overlaps with that of Gdf5 in the interzones of the developing joints (Fig. 1a). By E14.5, Gdf5 expression is similarly restricted to a subset of Creb5-expressing cells in the forelimb autopod (Fig. 1b). Consistent with these findings, Creb5 protein is specifically expressed in the developing shoulder, elbow, radiocarpal and ulnocarpal interzones at E14.5, and is subsequently expressed in both the carpal joints and metacarpophalangeal joints at E18.5 (Fig. 1c). While in the developing long bones Creb5 is only expressed in the articular perichondrium (Fig. 1c, yellow arrows) and not in the metaphyseal/diaphyseal perichondrium (Fig. 1c, white arrows); in the carpal and tarsal cartilage elements (i.e., the mesopodial skeleton) Creb5 is expressed in all the mesopodial perichondrium (Fig. 1c). Thus, the expression of Creb5 in both the articular and mesopodial perichondrium perfectly correlates with the fate of these regions of the perichondrium to give rise to articular cartilage[19]. In contrast the Creb5-negative metaphyseal/diaphyseal perichondrium develops into cortical bone[6,7].

To determine the role of Creb5 in joint formation, we employed CRISPR/Cas9 to produce a mouse line containing an in frame deletion of exon 9 in one allele of *Creb5* (*Creb5*[Δ9/+] mice; Supplementary Fig. 1a–d). This exon encodes both the basic region and 3 out of 5 leucines of the Creb5 bZIP domain, which encodes the DNA binding domain of Creb5[20] and is critically required for both dimerization and DNA interaction of bZIP transcription factors[21]. While infection of deep zone bovine articular chondrocytes with lentivirus encoding either Creb5(WT)-HA or a mutant form of Creb5 lacking the amino acids encoded by exon 9 (i.e., Creb5(Δ9)-HA) drove expression of nearly equal levels of HA-tagged protein (Supplementary Fig. 1e), Chromatin-Immunoprecipitation (ChIP)-PCR revealed that only Creb5(WT)-HA (and not Creb5(Δ9)-HA) was able to bind to the enhancers that drive Prg4 expression (Supplementary Fig. 1f). Consistent with the inability of Creb5(Δ9)-HA to bind to the Prg4 enhancers, we found that while Creb5(WT)-HA was able to induce robust expression of Prg4 in deep zone bovine articular chondrocytes treated with TGF-β2 and TGFα, Creb5(Δ9)-HA was unable to induce Prg4 expression in these cells (Supplementary Fig. 1g). Taken together, these results indicate that the *Creb5*[Δ9] allele encodes a mutant protein (i.e., Creb5(Δ9)) which is both unable to bind to enhancers or induce expression of transcriptional targets via direct protein: DNA interaction.

*Creb5*[Δ9/+] mice were viable, fertile, and displayed no readily apparent skeletal defects. In contrast, like *Creb5*[−/−] mice[22], *Creb5*[Δ9/Δ9] mice died immediately after birth and exhibited a slight shortening of their long bones (Supplementary Fig. 2). Gdf5, Wnt4, Wnt9a, and Wnt16 have all been found to play important roles in the early steps of joint formation[23–27]. Notably, while both Gdf5 and Wnt4 were robustly expressed in the interzones of the forming metacarpophalangeal joints in control E13.5 embryos (Fig. 2a, arrows), expression of these interzone markers was either reduced (Gdf5) or undetectable (Wnt4) in comparable regions of E13.5 *Creb5*[Δ9/Δ9] forelimb autopods (Fig. 2a). A Gdf5-expressing interzone was similarly absent from the developing tibiofemoral joint (in the knee), which failed to separate into distinct cartilage elements in E14.5 *Creb5*[Δ9/Δ9] embryos (Fig. 2b). We also noted that expression of both Gdf5 and Wif1, a negative regulator of Wnt-signaling[28] that is expressed in the developing joint interzones and articular cartilage[29], were each profoundly depressed in the anlage of either the radiohumeral joint (in the elbow) or the glenohumeral joint (in the shoulder) that had also failed to separate in E14.5 *Creb5*[Δ9/Δ9] embryos (Fig. 2c). Notably, the absence of an interzone between cartilage elements correlated with a loss of expression of Wnt4, Wnt9a, and Wnt11 specifically between the fused joints of *Creb5*[Δ9/Δ9] embryos (Fig. 2a, e, f); and is consistent with prior work indicating the necessity for canonical Wnt signaling in the generation of synovial joints[23–25,30]. Interestingly, while Wnt4 expression was lost from joint interzones between the fused cartilage elements, Wnt4 was still expressed in perichondrium located "adjacent" to the developing tibiofemoral joint in both *WT* and *Creb5*[Δ9/Δ9] embryos (Fig. 2d). In contrast to the radio-humeral, glenohumeral, and tibiofemoral joints that failed to express interzone markers in the absence of Creb5 function, the expression of interzone markers and joint separation were both observed in the ulnohumeral joint in *Creb5*[Δ9/Δ9] embryos (Fig. 2a, arrowheads). Taken together, these results indicate that Creb5 is critical to maintain appropriate expression of Gdf5, several Wnt family members and Wif1 in the developing interzones of most (but not all) synovial joints.

### *Creb5* is necessary to form mouse synovial joints

Consistent with defective joint interzone formation during embryogenesis, the femur and tibia failed to separate, and the entire knee joint (including cruciate ligaments, articular cartilage, and meniscus)

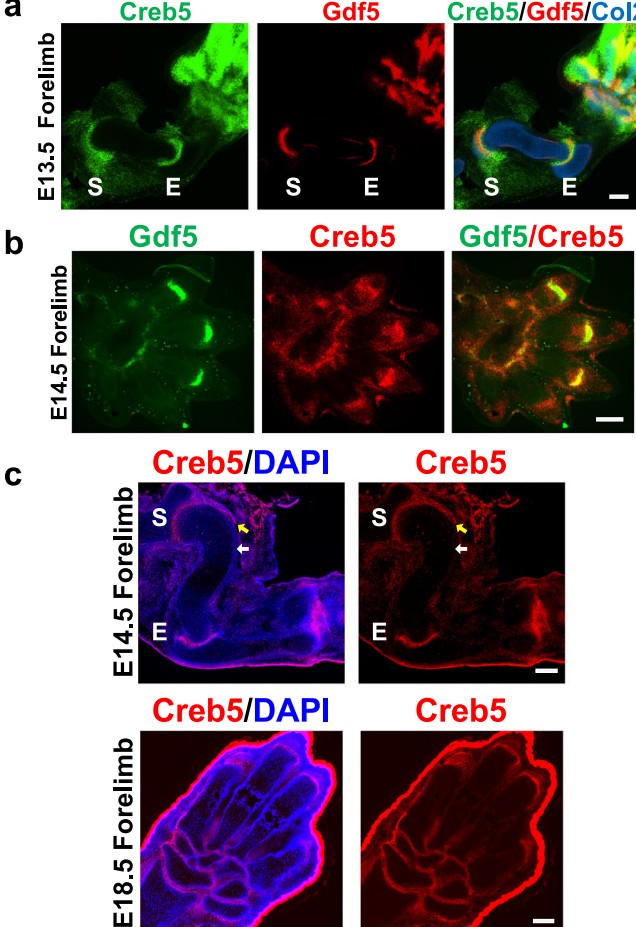

**Fig. 1 | Creb5 and Gdf5 expression overlap in the developing synovial joint interzones. a, b** Gene expression in E13.5 and E14.5 forelimbs from WT mice assayed by multiplexed RNAscope FISH. **c** Immunofluorescent staining of E14.5 and E18.5 forelimbs from WT mice for Creb5 and DAPI (to visualize nuclei). The articular perichondrium (yellow arrows) and metaphyseal/diaphyseal perichondrium (white arrows) are indicated. Shoulder (S) and Elbow (E) indicated. Similar results were obtained in at least two independent biological repeats. Scale bar equals 200 microns.

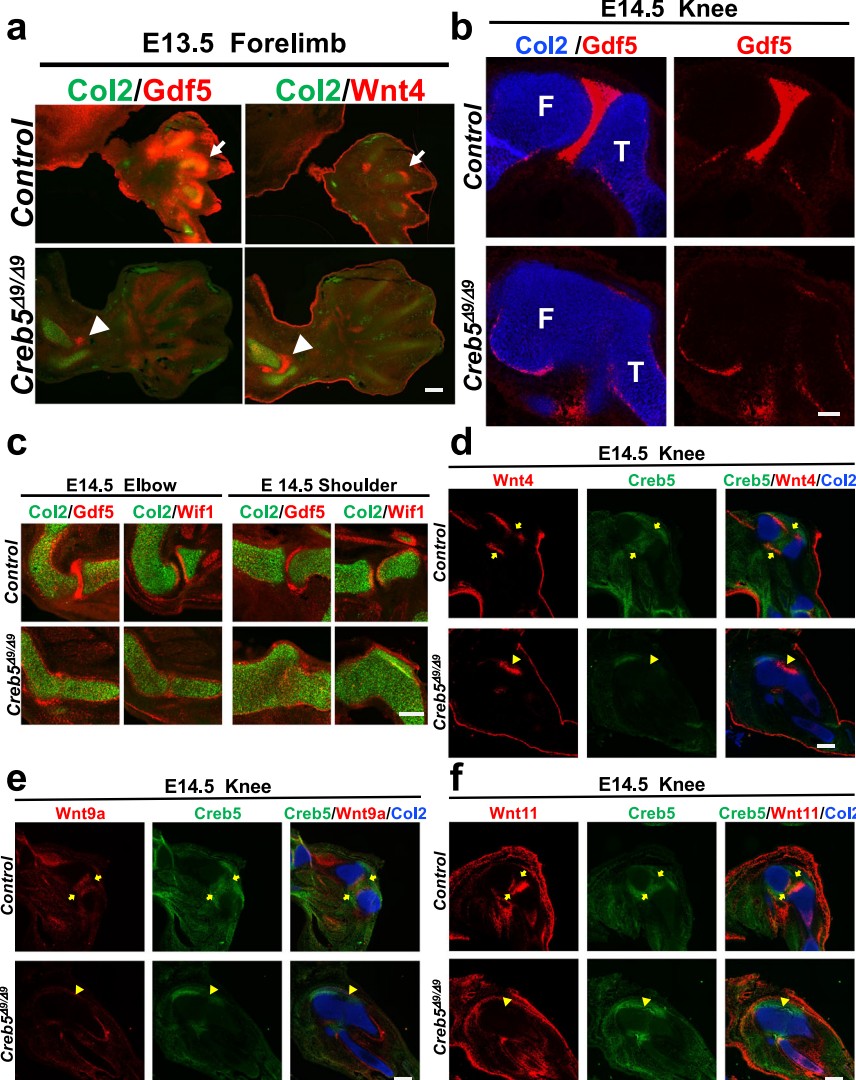

**Fig. 2 | Creb5 is necessary for synovial joint interzone formation. a** Gene expression assayed by FISH in serial sections of the forelimb of an E13.5 *Creb5^{Δ9/Δ9}* embryo or a control littermate (which is either *Creb5^{Δ9/+}* or *Creb5^{+/+}*). Arrows indicate the interzone of the forming metacarpophalangeal joints in the control embryo. Arrowheads indicate the interzone of the ulnohumeral joint in a *Creb5^{Δ9/Δ9}* embryo. **b** Gene expression assayed by RNAscope FISH in the hindlimb of an E14.5 *Creb5^{Δ9/Δ9}* embryo or control littermate. Femur (F) and Tibia (T) are indicated. Scale bar equals 100 microns. **c** Gene expression assayed by FISH in serial sections of the forelimb from an E14.5 *Creb5^{Δ9/Δ9}* embryo or control littermate. **d–f** Gene expression assayed by RNAscope FISH in serial sections of the knee of an E14.5 *Creb5^{Δ9/Δ9}* embryo or control littermate. In control limbs, the location of the articular perichondrium between the separated cartilage elements (yellow arrows) is indicated. In *Creb5^{Δ9/Δ9}* mutant limbs, the junction of the fused cartilage elements (yellow arrowheads) is indicated. Similar results were obtained in at least two independent biological repeats. Scale bar equals 200 microns, unless indicated otherwise.

was absent in P0 *Creb5^{Δ9/Δ9}* mice (14 out of 14 mice) (Fig. 3a, b). In addition, in P0 *Creb5^{Δ9/Δ9}* mice the distal epiphysis of the humerus was reduced in size and had failed to separate from the radius (Fig. 3c) (14 out of 14 mice); the pectoral girdle had failed to separate from the humerus (6 out of 6 mice) (Supplementary Fig. 3a); and the pelvic girdle had failed to separate from the femur (4 out of 4 mice) (Supplementary Fig. 3b). In the autopods, *Creb5^{Δ9/Δ9}* neonatal mice were also missing the radiocarpal, ulnocarpal, metacarpophalangeal, metatarsophalangeal, and interphalangeal joints (14 out of 14 mice) (Fig. 3a). Notably, Prg4-expressing cells were absent in both the fused tibiofemoral cartilage element (Fig. 3b) and the fused acetabular-femoral head cartilage element (Supplementary Fig. 3b) in the knees and hips of *Creb5^{Δ9/Δ9}* mice. Interestingly however, Prg4 expression was low but detectable in the elbows of these animals; and was expressed by cells located either at the surface of the remaining ulnohumeral joint or buried within the Col2a1-expressing core of the fused radiohumeral and glenohumeral cartilage elements (Fig. 3c and Supplementary

Fig. 3a). Immunocytochemistry for Creb5 revealed that Creb5(Δ9) protein was expressed in the perichondrium surrounding the radio-humeral fusion in *Creb5^{Δ9/Δ9}* mice (Fig. 3c), suggesting that Creb5 DNA binding is critical for both separation of the radius and humerus and for appropriate localization of Prg4+ cells in the elbow. Consistent with the requirement for TGF-β signaling to maintain Prg4 expression in articular cartilage[31,32], we noted that carboxy-terminal phosphorylated-Smad2 (which is a proxy for active TGF-β signaling) was also detectable in perichondrium of either control mice or *Creb5^{Δ9/Δ9}* mice (Fig. 3c).

While articular chondrocytes express Col2a1 but not Matrilin1 (Matn1), epiphyseal chondrocytes (which will later undergo endo-chondral ossification) express both Matn1 and Col2a1[33] (Fig. 3b and Supplementary Fig. 3). Notably, both Col2a1 and Matn1 were expressed throughout the entire fused tibiofemoral cartilage element in *Creb5^{Δ9/Δ9}* mice (Fig. 3b), revealing a profound absence of articular cartilage in the knees of these animals. In contrast to the knees, Matn1 expression was excluded from cells in both the presumptive shoulder (Supplementary

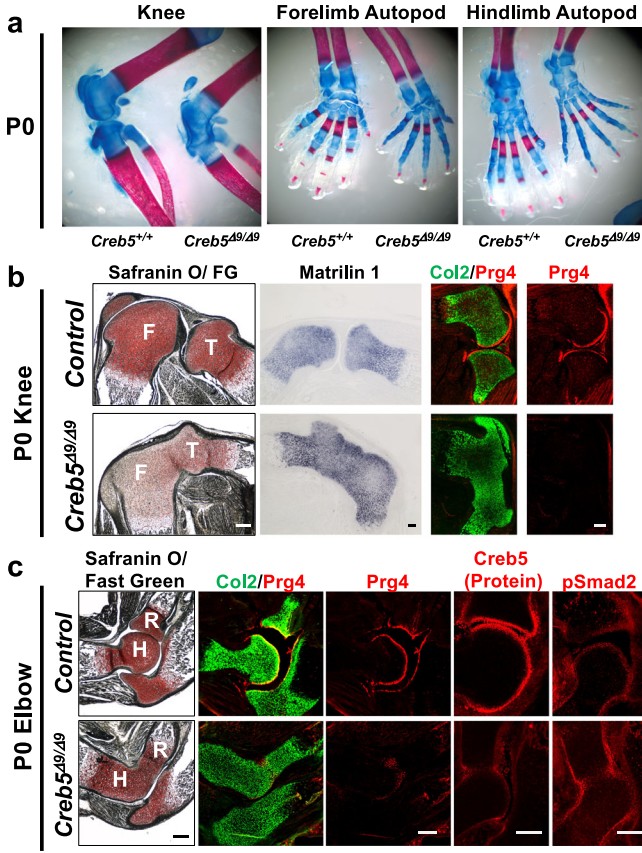

**Fig. 3 | Creb5 is necessary for the formation of many synovial joints. a** Alcian Blue/Alizarin Red staining of the knees and autopods of P0 *Creb5^Δ9/Δ9* mice or their WT littermates. Gene expression assayed by FISH and immunocytochemistry for Creb5 and carboxy-terminal phosphorylated Smad2 (pSmad2) in sections of the knees (**b**) and elbows (**c**) of P0 *Creb5^Δ9/Δ9* mice or their control littermates. Parallel sections were stained with Safranin O/Fast Green (FG). Femur (F), Tibia (T), Humerus (H) and Radius (R) are indicated. Similar results were obtained in either 14 (**a**), 4 (**b**) or 2 (**c**) independent biological repeats. Scale bar equals 200 microns.

Fig. 3a) and presumptive hip (Supplementary Fig. 3b) joints. In contrast to the stylopod, zeugopod, and autopod cartilage elements which displayed failures in joint separation in the absence of Creb5, most (but not all) carpal and tarsal cartilage elements continued to separate in *Creb5^Δ9/Δ9* mice and contained an outer layer of Matn1-negative cells (Supplementary Fig. 3c).

## Reciprocal expression of *Creb5* and *Wnt5a* in the perichondrium

We noted that the expression of Creb5 and Wnt5a were mutually exclusive in the perichondrium surrounding forming cartilage elements in the stylopod and zeugopod, with expression of Wnt5a being absent from the articular (epiphyseal) perichondrium and thus restricted to the metaphyseal and diaphyseal perichondrium (Fig. 4a, b, white arrows). In contrast, Creb5 was specifically expressed in the articular perichondrium (Fig. 4a, b, yellow arrows) in both the developing elbow and knee joints. While E14.5 forelimb autopods contained distal regions with overlapping expression of Creb5 and Wnt5a, the perichondrium of the carpal elements in these developing limbs expressed Creb5 but lacked detectable expression of Wnt5a (Fig. 4a). Thus, as synovial joints are formed, the perichondrium is divided into regions with either high level expression of Creb5 (which mark the joint interzone) versus regions with high level expression of Wnt5a (which lie adjacent to the developing growth plates). Interestingly, in *Creb5^Δ9/Δ9* embryos, expression of *Wnt5a* encroached into the perichondrium located adjacent to the fused tibiofemoral cartilage element and was even expressed in perichondrial cells expressing *Creb5^Δ9* transcripts

(Fig. 4b, white arrowheads). Thus, not only is Creb5 activity necessary to promote the expression of joint interzone markers, but it is also critical to exclude the expression of *Wnt5a* from perichondrium adjacent to the forming joint.

## Ectopic expression of *Creb5* blocks that of *Wnt5a* in perichondrium

Because Creb5 is specifically expressed in articular perichondrium (which surrounds the epiphysis) but not in Wnt5a-expressing metaphyseal/diaphyseal perichondrium, we asked whether misexpression of Creb5 in limb bud mesenchymal cells would either alter the expression of Wnt5a and/or affect growth of long bones. To this end, we targeted a Cre-inducible, HA epitope-tagged bovine Creb5 cDNA, upstream of ires-GFP, into the *Rosa26* locus (*Rosa26^iCreb5-HA*, diagrammed in Supplementary Fig. 4a). As expected, a quarter of the offspring from crosses between *Prx1-Cre*[34] and *Rosa26^iCreb5-HA* mice (i.e., *Prx1-Cre-iCreb5* mice) expressed iCreb5-HA^(Prx1-Cre) in their limb bud mesenchymal cells (Supplementary Fig. 4c). While both the Creb5-HA and GFP transgenes were expressed throughout the limb bud mesenchyme in E11.5 *Prx1-Cre-iCreb5* embryos (Supplementary Fig. 5a, b), transgene-expressing cells were excluded from the cartilage elements and were restricted to the perichondrium in the limb buds of E14.5 *Prx1-Cre-iCreb5* embryos (Fig. 5a, b). RNAscope FISH, employing probes that detect both exogenous (bovine) and endogenous (mouse) Creb5, Wnt5a, and Col2a1 indicated that a Creb5+/Wnt5a- perichondrium, that is normally restricted to the joint forming regions (yellow arrows) and is absent from the metaphyseal/diaphyseal regions (white arrows), has been extended to lie adjacent to the metaphyseal/diaphyseal regions of the forming zeugopod cartilage elements in E14.5 *Prx1-Cre-iCreb5* embryos (Fig. 5a, yellow arrowheads). Notably, iCreb5-HA^(Prx1-Cre)-expressing cells were absent from the Col2a1-expressing cartilage core and restricted to perichondrial tissues lacking Wnt5a expression (Fig. 5a, b); suggesting that iCreb5-HA^(Prx1-Cre) expression directs cells to remain in the perichondrium. Expression of cellular retinoic acid binding protein-1 (Crabp1) which, like Wnt5a, is expressed in the metaphyseal/diaphyseal perichondrium[35] (and is excluded from the articular perichondrium) was also absent from perichondrial cells programmed to express iCreb5-HA^(Prx1-Cre) (Fig. 5b). Consistent with the requirement for Wnt5a to induce expression of the osteogenic transcription factor Runx2 in the developing periosteum[36], Runx2 expression was profoundly decreased in the perichondrium surrounding the stylopod and zeugopod cartilage elements of E14.5 *Prx1-Cre-iCreb5* embryos (Fig. 5c). Notably, the iCreb5-HA^(Prx1-Cre)-expressing perichondrium displayed elevated and ectopic expression of the joint interzone markers Gdf5, Wnt11 and Wnt4 (Fig. 5b, yellow arrowheads and Supplementary Fig. 6, yellow arrowheads), which usually display nested patterns of expression in the articular perichondrium (Fig. 5b, yellow arrows and Supplementary Fig. 6, yellow arrows).

Consistent with the role for *Wnt5a* to support both chondrocyte hypertrophy and longitudinal bone growth[36,37], both the stylopod and zeugopod cartilage elements in E14.5 *Prx1-Cre-iCreb5* limb buds lacked the expression of Col10a1 (Fig. 5b); and many proximal forelimb and hindlimb cartilage elements in newborn *Prx1-Cre-iCreb5* mice failed to grow longitudinally (7 out of 7 mice) (Fig. 6a–c). Except for parts of the autopod, most limb cartilage elements lacked cortical bone and any columnar organization of a growth plate; and consisted of randomly oriented chondrocytes that resembled the resting zone of the normal epiphysis (Fig. 6c). In addition to a blockade of longitudinal growth in both the stylopod and zeugopod, the hindlimb autopods of P0 *Prx1-Cre-iCreb5* mice displayed polydactyly and an excessive number of tarsal elements (Fig. 6a). In addition to inducing hindlimb polydactyly, we noted that both forelimb and hindlimb buds were larger in E11.5 *Prx1-Cre-iCreb5* embryos versus their control littermates (Supplementary Fig. 5a). To understand what might cause these phenotypes, we

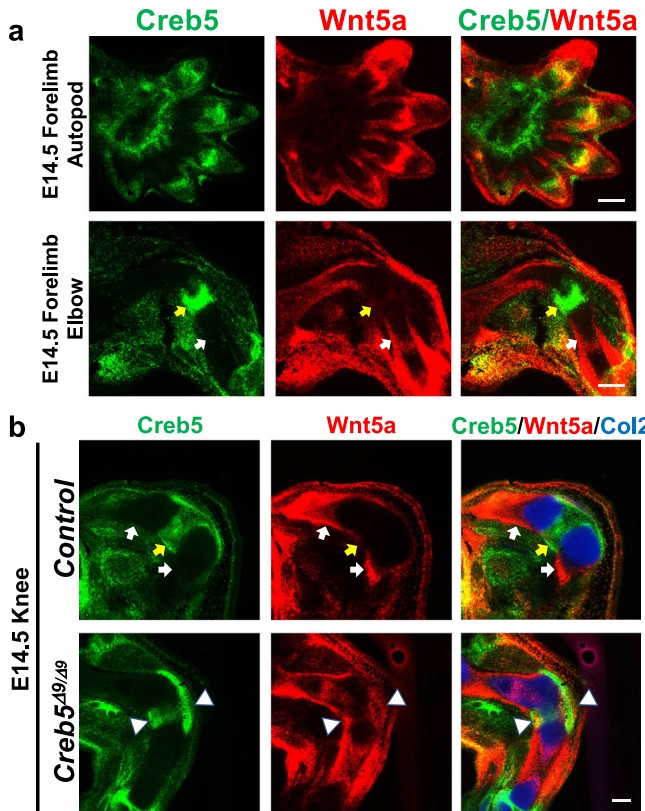

**Fig. 4 | Creb5 is expressed in a reciprocal pattern to Wnt5a in the perichondrium.** Gene expression assayed by RNAscope FISH in the forelimb of a WT E14.5 mouse embryo (**a**) or the knee of an E14.5 $Creb5^{\Delta9/\Delta9}$ embryo or control littermate (**b**). In WT/control limbs, the articular perichondrium (yellow arrows) and metaphyseal/diaphyseal perichondrium (white arrows) are indicated. In $Creb5^{\Delta9/\Delta9}$ mutant limb, the perichondrial cells expressing both Wnt5a and mutant $Creb5^{\Delta9/\Delta9}$ transcripts (white arrowheads) are indicated. Similar results were obtained in at least three independent biological repeats. Scale bar equals 200 microns.

examined the expression of Msx1 in the hindlimb buds of E11.5 *Prx1-Cre-iCreb5* embryos and control littermates. Prior work has established that BMP signaling promotes the expression of Msx1 in the limb bud mesenchyme[38,39], can drive apoptosis of limb bud mesenchymal cells[40–44], and also restricts the expression of Fgf4/8 in the apical ectodermal ridge (AER)[39]. Indeed, manipulations that decrease the level of BMP signaling in the limb bud both promote an expansion of Fgf4/8 (and the AER) into a larger region of the limb bud ectoderm[39,45] and induce polydactyly[45–47]. Interestingly, we observed that misexpression of iCreb5-HA[(Prx1-Cre)] in E11.5 embryos both expanded the expression of Gdf5 to a broader domain of the limb bud mesenchyme; and simultaneously decreased the expression of Msx1 in the limb bud (Supplementary Fig. 5c). As Gdf5 has been found to block signaling by BMPR1a[48] (reviewed in ref. 49), we speculate that both the increased size of limb buds in E11.5 *Prx1-Cre-iCreb5* embryos, repression of Msx1-expression, and hindlimb polydactyly may all reflect an attenuation of BMP signaling in the limb bud mesenchyme due to ectopic expression of Gdf5.

## iCreb5-HA[(Prx1-Cre)] disrupts a Wnt5a-positive feedback-loop

Prior work has indicated that maintained perichondrial expression of Wnt5a in the developing zeugopod cartilage elements requires functional Wnt5a ligand[36]. We confirmed this result, by examining expression of Wnt5a in embryos containing a deletion of exon 2 in both alleles of $Wnt5a$ ($Wnt5a^{\Delta exon2/\Delta exon2}$) and employing a Wnt5a RNAscope probe outside of the deleted exon. Notably, perichondrial expression of Wnt5a is extinguished, and expression of its downstream target gene Ctgf is

significantly reduced, in the zeugopod cartilage elements of E14.5 $Wnt5a^{\Delta exon2/\Delta exon2}$ embryos (Fig. 6d). In contrast, expression of Wnt5a is still maintained in the distal autopod region of E14.5 $Wnt5a^{\Delta exon2/\Delta exon2}$ embryos (Fig. 6d). These findings indicate that Wnt5a function is required to maintain its own expression specifically in the perichondrium. Indeed, Guan and colleagues have previously found that Wnt5a can autoregulate its own expression via a Yap/Taz/Tead signaling pathway in cultured cells[50]. Thus, it is plausible that a Wnt5a positive feedback-loop similarly maintains its own expression in the perichondrium of developing long bones; and is necessary to up-regulate the expression of both itself and other Yap/Taz target genes (i.e., Ctgf) in the adjacent cartilage element. Interestingly, the cartilage elements in *Prx1-Cre-iCreb5* embryos which failed to elongate similarly lacked perichondrial expression of Wnt5a; and displayed reduced Ctgf expression in the adjacent cartilage tissue (Fig. 6e), suggesting that ectopic perichondrial expression of iCreb5-HA[(Prx1-Cre)] somehow disrupts a Wnt5a-positive feedback-loop.

## iCreb5-HA[(Prx1-Cre)] promotes generation of an articular perichondrium

iCreb5-HA[(Prx1-Cre)]-expressing cells in *Prx1-Cre-iCreb5* mice became localized to perichondrial tissues that specifically lack the expression of Wnt5a, as does the articular perichondrium. Thus, we wondered whether iCreb5-HA[(Prx1-Cre)]-expressing cells would develop into articular chondrocyte progenitors, that express both Prg4 and Wif1. Indeed, in P0 *Prx1-Cre-iCreb5* mice, the stylopod, zeugopod and tarsal elements were encased by a peripheral ring of cells which expressed the articular cartilage markers Prg4[11] and Wif1[29] (Fig. 7a). Beneath the superficial layer of Creb5[+]/Prg4[+]/Wif1[+] cells was a deeper zone of chondrocytes which expressed Col2a1 but that lacked expression of Matn1 in the limbs of *Prx1-Cre-iCreb5* mice (Fig. 7a), thus resembling articular cartilage[33]. Our findings suggest that iCreb5-HA[(Prx1-Cre)] expression in limb bud mesenchymal cells both promotes the generation of an articular perichondrium and blocks formation of both cortical bone and growth plates within the forming cartilage elements.

To determine whether the profound inhibition of longitudinal growth of the stylopod and zeugopod in *Prx1-Cre-iCreb5* mice (Fig. 7b, c) was due to iCreb5-HA[(Prx1-Cre)]-expression in either Sox9[+] chondrocytes or in other tissues that surround developing cartilage elements, we mated $Sox9^{ires-Cre}$ mice[51] with $Rosa26^{iCreb5-HA}$ mice to generate *Sox9-ires-Cre-iCreb5* embryos, which expressed iCreb5-HA[(Sox9-Cre)] in most chondrocytes and in a few perichondrial cells (Fig. 7d, e). In striking contrast to the absence of both appendicular growth plates and chondrocyte maturation following misexpression of iCreb5-HA[(Prx1-Cre)] in limb bud mesenchymal cells (in *Prx1-Cre-iCreb5* mice); appendicular growth plates formed and chondrocyte maturation was only slightly delayed following misexpression of iCreb5-HA[(Sox9-Cre)] specifically in chondrocytes (in *Sox9-ires-Cre-iCreb5* embryos) (Fig. 7d, e and Supplementary Fig. 7). Notably, while iCreb5-HA[(Prx1-Cre)] was robustly expressed in much of the perichondrium in the limbs of *Prx1-Cre-iCreb5* mice (Fig. 7c), iCreb5-HA[(Sox9-Cre)] was only expressed in a small minority of perichondrial cells in *Sox9-ires-Cre-iCreb5* embryos (Fig. 7e). Thus, the extent of expression of iCreb5 in the perichondrium correlates with the ability of this transcription factor to block both growth plate formation and chondrocyte maturation (diagrammed in Fig. 7f).

## Wif1 and Fat4 are transcriptional targets of Creb5

To further elucidate how Creb5 might regulate Wnt5a expression we performed an unbiased screen to identify Creb5 target genes. RNA-Seq analysis revealed that shRNA-mediated knockdown of Creb5 in superficial zone bovine articular chondrocytes reduced the expression of several classes of signaling molecules, including Wif1, Gdf10, Fat1, Fat4, EphA3, Hbegf (an EGFR-ligand), and Tgfbr1 (Supplementary Fig. 8a, b and Supplementary Data 1 and 2). Wif1 has been noted

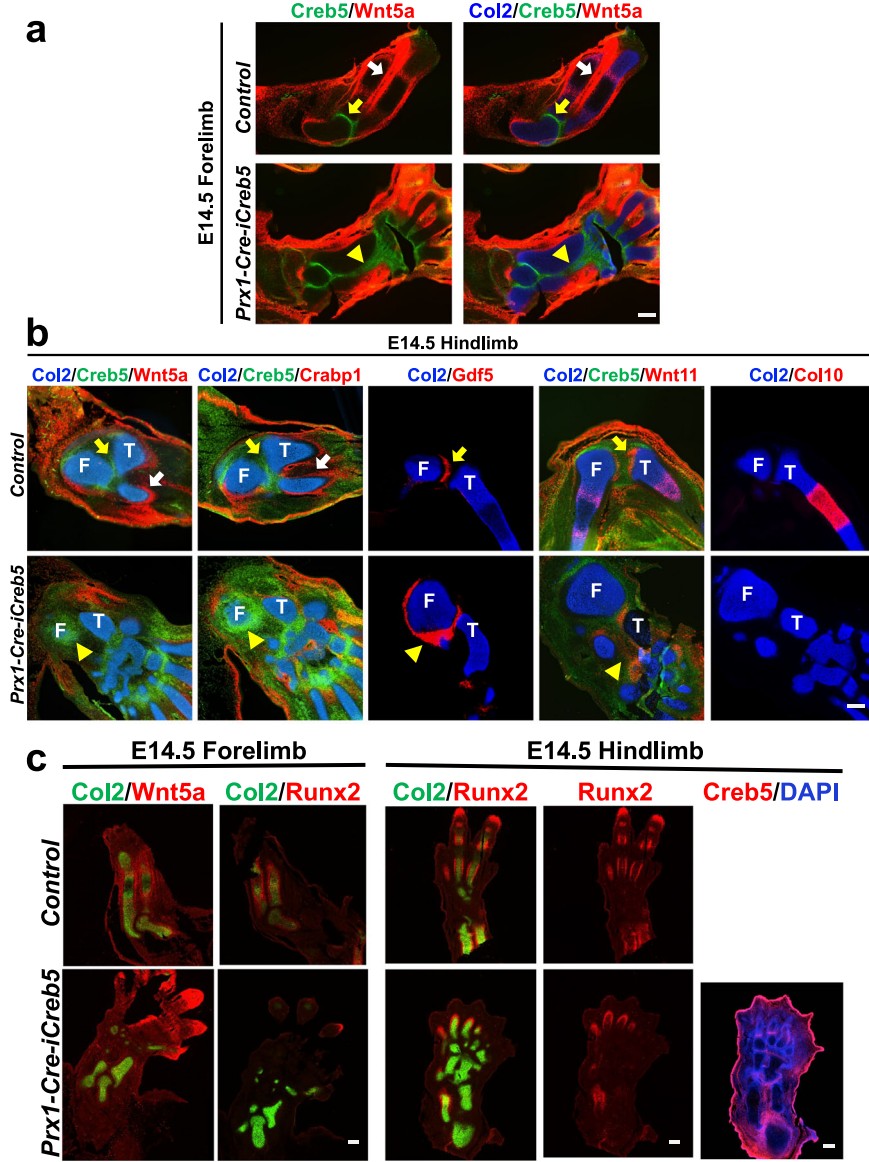

**Fig. 5 | Misexpression of iCreb5-HA^(Prx1-Cre) inhibits perichondrial expression of Wnt5a, Crabp1 and Runx2.** Gene expression assayed by RNAscope FISH (**a**, **b**) or FISH (**c**) in serial sections of indicated limbs of either E14.5 *Prx1-Cre-iCreb5* embryos or control littermates. In control limbs, the articular perichondrium (yellow arrows) and metaphyseal/diaphyseal perichondrium (white arrows) are indicated. In *Prx1-Cre-iCreb5* limbs, the iCreb5-HA^(Prx1-Cre)+, Wnt5a⁻/Crabp1⁻ perichondrium (yellow arrowheads) is indicated. Femur (F) and Tibia (T) are indicated. Right-most panel of **c** displays immunofluorescent staining for Creb5 and DAPI in serial section of the same hindlimb analyzed by FISH. Red fluorescent signal in ectoderm is non-specific. Similar results were obtained in either 5 (**a**) or 2 (**b**, **c**) independent biological repeats. Scale bar equals 200 microns.

to both bind to Wnt5a[29] and inhibit Wnt5A-induced actin polymerization[52], which in turn promotes Yap/Taz signaling[53,54]. In addition, interaction of the atypical cadherins Fat1 and Fat4 with their binding partners (Dchs1/2) has also been noted to block expression of Yap/Taz/Tead target genes[55–57]. Consistent with the notion that Creb5 promotes the expression of both Wif1 and Fat4, we noted that both these Creb5 target genes are expressed in the interzone of developing synovial joints and that their expression is lost in these tissues in *Creb5^Δ9/Δ9* mice (Figs. 2c and 8a). While perichondrial expression of Wif1 is restricted to the articular perichondrium (Fig. 2c), Fat4 is expressed at high levels in articular perichondrium which lacks Wnt5a expression (Fig. 8a–c, yellow arrows); and expressed at relatively lower levels in perichondrium which expresses Wnt5a and lies adjacent to the primary ossification center (Fig. 8a–c, white arrows). Notably, ectopic perichondrial expression of iCreb5-HA^(Prx1-Cre) in *Prx1-Cre-iCreb5* embryos promotes

high level expression of both Wif1 (Fig. 7a) and Fat4 (Fig. 8b, c) in the perichondrium that encases the zeugopod cartilage elements that lack growth plates; and simultaneously blocks expression of Wnt5a in this same tissue (Fig. 8c, yellow arrowheads). Taken together, our findings indicate that Creb5 is necessary to drive the expression of several signaling molecules in the joint interzone, including Gdf5, Wnt9a, Wnt11, Wif1 and Fat4; and/or maintains their expression by establishing the interzone cell fate.

**Creb5 maintains zonal gene expression in articular cartilage**

To study the role of Creb5 in postnatal synovial joints we generated mice containing Creb5 alleles in which exon 9 (which encodes the bZIP DNA binding domain) is flanked by loxP sites (*Creb5^flox9/flox9* mice; Supplementary Fig. 9a, b). Like *Creb5^Δ9/Δ9* animals, *Prx1-Cre; Creb5^flox9/flox9* mice also displayed an absence of a knee joint (Supplementary Fig. 9c), indicating that the floxed alleles in *Creb5^flox9/flox9* mice can be deleted by

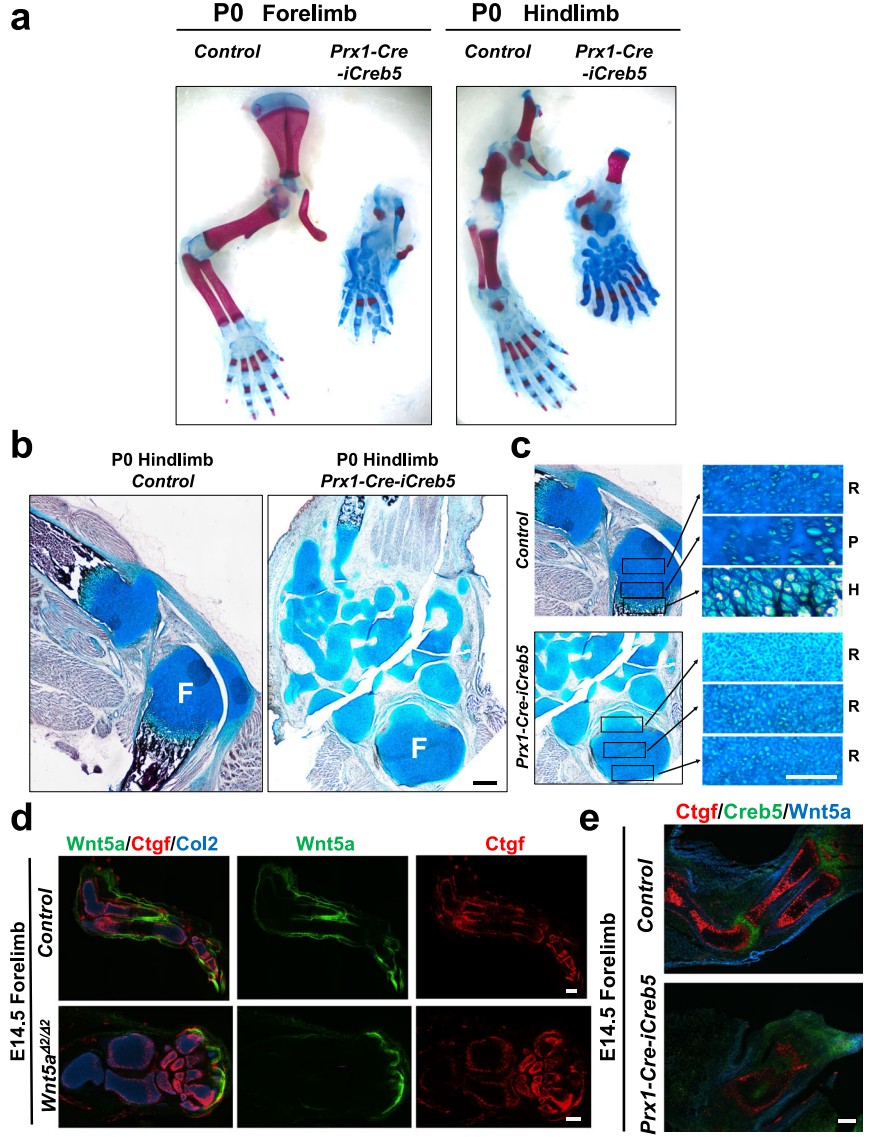

**Fig. 6 | Misexpression of iCreb5-HA<sup>(Prx1-Cre)</sup> blocks growth plate formation and phenocopies aspects of Wnt5a loss of function. a** Alcian Blue/Alizarin Red whole mount staining and **b, c** Von Kossa/Alcian Blue staining of indicated limbs from either P0 *Prx1-Cre-iCreb5* mice or control littermates. Femur (F), Resting zone chondrocytes (R), Proliferating zone chondrocytes (P) and Hypertrophic zone chondrocytes (H) are indicated. Scale bar equals 100 microns in **c. d**, **e** Gene expression assayed by RNAscope FISH in E14.5 forelimbs of either control littermates, *Wnt5aΔexon2/Δexon2* embryos or *Prx1-Cre-iCreb5* embryos. Similar results were obtained in either 7 (**a**), 2 (**b, c, e**) or 3 (**d**) independent biological repeats. Scale bar equals 200 microns, unless otherwise indicated.

Cre-mediated recombination. We mated *Creb5<sup>flox9/flox9</sup>* mice with *Aggrecan1<sup>tm(IRES-CreERT2)</sup>;Creb5<sup>flox9/flox9</sup>* mice to generate litters containing both *Aggrecan1<sup>tm(IRES-CreERT2)</sup>; Creb5<sup>flox9/flox9</sup>* mice or *Creb5<sup>flox9/flox9</sup>* control mice (which lack the CreERt2 driver allele). Tamoxifen was repeatedly administered to these litters at postnatal days 1–11, to specifically delete exon 9 of *Creb5<sup>flox9/flox9</sup>* in postnatal stage articular chondrocytes. Following sacrifice at P14, RT-qPCR analysis of RNA isolated from the femoral heads of *Aggrecan1<sup>tm(IRES-CreERT2)</sup>; Creb5<sup>flox9/flox9</sup>* mice or *Creb5<sup>flox9/flox9</sup>* control mice indicated that exon9 of *Creb5<sup>flox9/flox9</sup>* was deleted in femoral head articular cartilage of the *Aggrecan1<sup>tm(IRES-CreERT2)</sup>; Creb5<sup>flox9/flox9</sup>* animals (Supplementary Fig. 9d). RNAScope analysis of gene expression in the knees of these animals revealed that expression of both Prg4 and Wif1 was markedly decreased in P14 *Aggrecan1<sup>tm(IRES-CreERT2)</sup>;Creb5<sup>flox9/flox9</sup>* mice in comparison to their control *Creb5<sup>flox9/flox9</sup>* littermates (Fig. 9a). In striking contrast to the loss of both Prg4 and Wif1 expression following deletion of *Creb5* in the forming joint, the expression of Ctgf was increased in both periarticular chondrocytes and in the metaphyseal perichondrium (Fig. 9a).

In addition to decreasing the expression of both Prg4 and Wif1 in the articular cartilage, we also noted that postnatal loss of Creb5 function resulted in a decreased number of Col10a1-expressing cells on the side of the secondary ossification center that is nearest the articular surface in P30 *Aggrecan1<sup>tm(IRES-CreERT2)</sup>;Creb5<sup>flox9/flox9</sup>* mice (Fig. 9b). This finding suggests that Creb5 (in the articular cartilage) induces the expression of signaling molecules that promote the maintenance of Col10a1-expressing cells in the adjacent secondary ossification center and blocks either their premature loss and/or developmental maturation into osteocytes. Consistent with the notion that Creb5 indirectly blocks premature endochondral ossification, we also observed premature hypertrophy (in P30 mice) and subsequent endochondral ossification (in P42 mice) of the anterior meniscus in *Aggrecan1<sup>tm(IRES-CreERT2)</sup>;Creb5<sup>flox9/flox9</sup>* mice that been treated with tamoxifen (Fig. 9c).

In parallel with the loss of Prg4 expression in the superficial cells of the articular cartilage in *Aggrecan1<sup>tm(IRES-CreERT2)</sup>;Creb5<sup>flox9/flox9</sup>* mice (Fig. 10a), we similarly observed that superficial zone flat cells on the

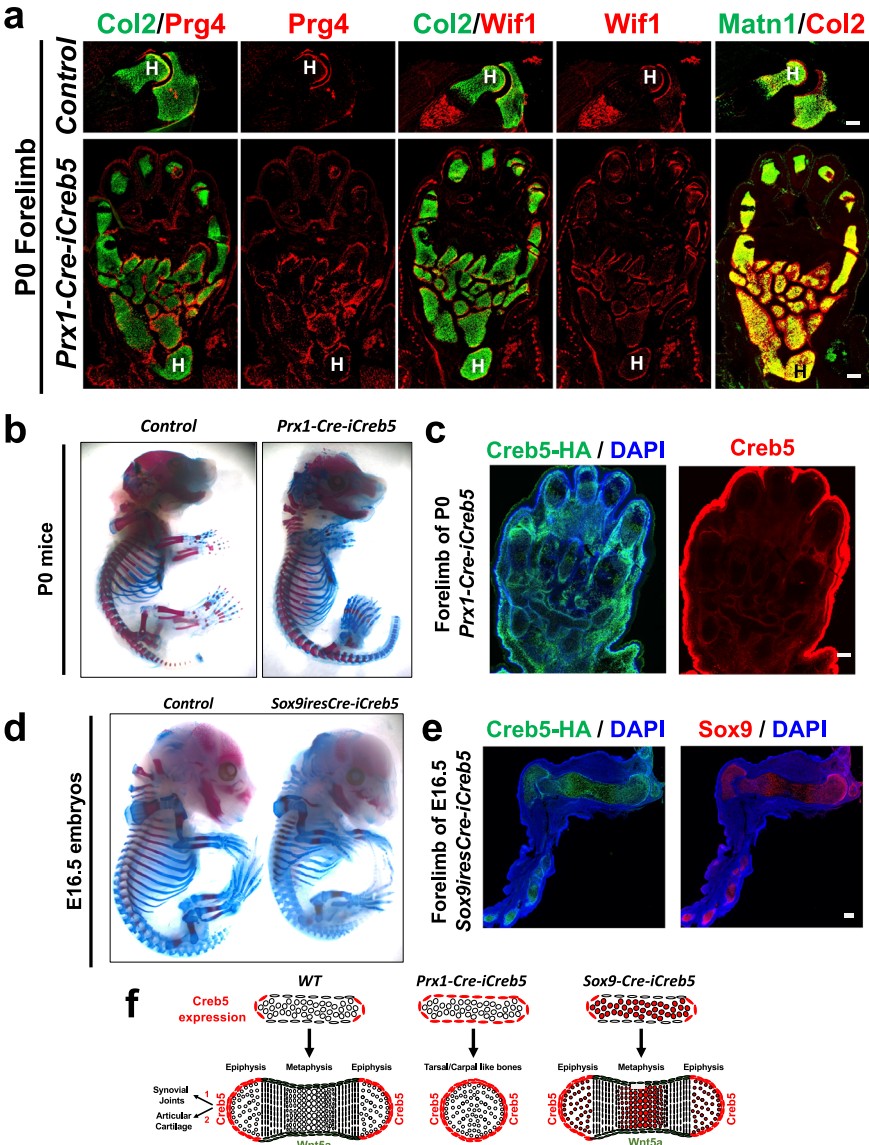

**Fig. 7 | Misexpression of Creb5 in all limb bud mesenchymal cells, but not in Sox9-expressing cells, profoundly blocks longitudinal growth in the stylopod and zeugopod cartilage elements and induces the perichondrium to develop as articular cartilage. a** Gene expression assayed by FISH in serial sections of forelimbs from either a P0 *Prx1-Cre-iCreb5* mouse or a control littermate. Humerus (H) is indicated. **b** Alcian Blue/Alizarin Red whole mount staining of either a P0 *Prx1-Cre-iCreb5* mouse or a control littermate (that lacks either the *Prx1-Cre* or the *Rosa26^iCreb5-HA* allele). **c** Immunofluorescence detection of either iCreb5-HA^(Prx1-Cre) (green), endogenous and exogenous Creb5 (red) and DAPI (to detect nuclei) in serial sections of the forelimb of a P0 *Prx1-Cre-iCreb5* mouse. Red fluorescent signal in ectoderm (in right-most image) is an artifact and was also observed in the absence of anti-Creb5. **d** Alcian Blue/Alizarin

Red whole mount staining of either an E16.5 *Sox9-ires-Cre-iCreb5* embryo or a control littermate (that lacks either the *Sox9-ires-Cre* or the *Rosa26^iCreb5-HA* allele).
**e** Immunofluorescence detection of iCreb5-HA^(Sox9-Cre), Sox9, and DAPI in the forelimb of an E16.5 *Sox9-ires-Cre-iCreb5* embryo. Similar results were obtained in either 5 (**a**), 7 (**b**), 2 (**c, e**) or 4 (**d**) independent biological repeats. Scale bar equals 200 microns.
**f** Our findings suggest that iCreb5-HA^(Prx1-Cre) expression in limb bud mesenchymal cells (other than in Sox9-expressing cells) promotes the generation of an articular perichondrium encasing the stylopod and zeugopod cartilage elements, blocks perichondrial expression of Wnt5a, Crabp1 and Runx2, and thus inhibits formation of both cortical bone and growth plates within these forming cartilage elements (which resemble tarsal/carpal-like bones).

surface of either the articular cartilage or the meniscus were largely absent from the knees of these animals (Figs. 10a and 9c). To quantify both the location and morphology of Col2a1-expressing cells in the articular cartilage of P30 *Aggrecan1^tm(IRES-CreERT2); Creb5^flox9/flox9* mice or *Creb5^flox9/flox9* control mice we performed RNAScope analysis (Fig. 10b). We noted that postnatal loss of Creb5 function in the articular cartilage led to approximately a 75% decline in the number of Col2a1-expressing flat cells in the superficial-most two cell layers of the articular cartilage, on both the femoral and tibial surface in the knees of P30 *Aggrecan1^tm(IRES-CreERT2); Creb5^flox9/flox9* mice (Fig. 10c).

## Discussion

Prior work has established that synovial joint progenitor cells that are marked by Gdf5 expression successively give rise to either epiphyseal chondrocytes (which eventually undergo endochondral ossification), articular chondrocytes (which maintain an immature chondrocyte phenotype), ligaments, meniscus, or synoviocytes that line the joint capsule[58–60]. Notably, the Gdf5-expressing synovial joint progenitors that specifically give rise to the articular cartilage express Prg4[3–5]. In this work, we document that the transcription factor Creb5 integrates several aspects of joint development: cueing the earliest stages of joint formation, modulating the morphology of the epiphysis, and

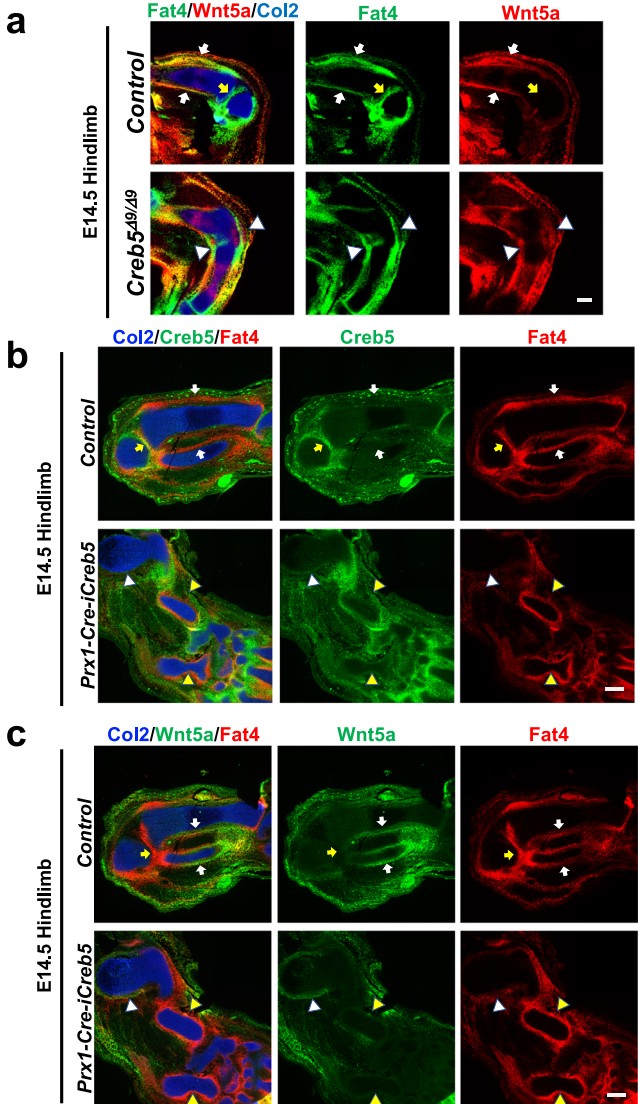

**Fig. 8 | Misexpression of iCreb5-HA<sup>(Prx1-Cre)</sup> promotes formation of a Fat4+, Wnt5a<sup>–</sup> perichondrium.** Gene expression assayed by RNAscope FISH in the hindlimbs of an E14.5 Creb5<sup>Δ9/Δ9</sup> embryo or control littermate (**a**) or in serial sections of the forelimbs of an E14.5 Prx1-Cre-iCreb5 embryo or control littermate (**b**, **c**). In control limbs, the Fat4<sup>high</sup>, Wnt5a<sup>low</sup> articular perichondrium (yellow arrows) and Fat4<sup>low</sup>, Wnt5a<sup>high</sup> metaphyseal/diaphyseal perichondrium (white arrows) are indicated. In Creb5<sup>Δ9/Δ9</sup> mutant limbs, the perichondrial cells expressing both Wnt5a and mutant Creb5<sup>Δ9/Δ9</sup> (white arrowheads) are indicated. In Prx1-Cre-iCreb5 limbs: Fat4<sup>low</sup>, Wnt5a<sup>high</sup> perichondrium (white arrowheads) or Fat4<sup>high</sup>, Wnt5a<sup>low</sup> perichondrium (yellow arrowheads) are indicated. In Prx1-Cre-iCreb5 embryos, perichondrial expression of iCreb5-HA<sup>(Prx1-Cre)</sup> promotes high level expression of Fat4 in the perichondrium, blocks expression of Wnt5a in this same tissue (yellow arrowheads) and inhibits chondrocyte hypertrophy. Perichondrium surrounding cartilage elements that display chondrocyte hypertrophy in Prx1-Cre-iCreb5 embryos continue to express WT levels of Wnt5a (white arrowheads). Similar results were obtained in two independent biological repeats. Scale bar equals 200 microns.

subsequently maintaining the expression of both Prg4 and other superficial zone-specific genes in the fully differentiated articular cartilage[18]. Creb5 is critical to maintain appropriate expression of both Wnt ligands that promote β-catenin stabilization (i.e., Wnt4, Wnt9a, Wnt11) and other signaling molecules (i.e., Gdf5 and Wif1) in the interzones of the developing synovial joints. Correlating with a loss of interzone-specific Wnt ligand expression, we have found that Col2a1 fails to be down-regulated in the non-separated regions of the fused joints in Creb5<sup>Δ9/Δ9</sup> mice. As Creb5 can bind to target sites as either a

homodimer or a heterodimer with Jun[20], which is critical to activate the expression of both Wnt9a and Wnt16 during the early stages of joint development[61], it is possible that Creb5 either directly or indirectly activates expression of its transcriptional targets in a complex with Jun. Indeed, we have found that interzone-specific expression of Wnt4, Wnt9a, Wnt11 and Gdf5 is highly dependent upon Creb5 function; and conversely that expression of Wnt4, Wnt11 and Gdf5 were each expanded in the perichondrium surrounding the stylopod and zeugopod cartilage elements of E14.5 Prx1-Cre-iCreb5 embryos. Prior work has indicated that the absence of skeletal muscle activity disrupts both the growth of bone eminences[62] and formation of a limited number of synovial joints; but notably does not affect formation of the knee joint[63]. In contrast, loss of Creb5 function does not affect the growth of the bone eminences but instead disrupts the formation of most synovial joints (including the knee). Thus, the absence of synovial joints following loss of Creb5 function cannot be explained by loss of skeletal muscle function; and is most consistent with the necessity for Creb5 in the joint interzone itself to drive the formation of these structures.

Creb5 is expressed in the articular perichondrium, which forms the articular cartilage[3–5], but is not expressed in the metaphyseal/diaphyseal perichondrium, that will give rise to the periosteal precursors of the cortical bone[6]. In contrast, both Wnt5a and Crabp1 are most highly expressed in the metaphyseal/diaphyseal perichondrium[35–37]. Thus, the expression domain of Creb5 versus that of Wnt5a/Crabp1 define two regions in the perichondrium, which give rise to distinct cell fates (i.e., articular cartilage and cortical bone, respectively). We have found that misexpression of iCreb5-HA<sup>(Prx1-Cre)</sup> in limb bud mesenchymal cells (driven by Prx1-Cre) both blocked perichondrial expression of Wnt5a and Crabp1, and simultaneously converted the metaphyseal/diaphyseal perichondrium of long bones (which usually develops into cortical bone) into articular perichondrium that expressed both Prg4 and Wif1. The ability of iCreb5-HA<sup>(Prx1-Cre)</sup> to promote perichondrial cells to develop into articular chondrocyte progenitors, versus periosteal progenitors, may be a consequence of the repression of Runx2 expression in the perichondrium. In addition to promoting the formation of articular perichondrium at the expense of periosteum in Prx1-Cre-iCreb5 mice, iCreb5-HA<sup>(Prx1-Cre)</sup>-induced loss of Wnt5a expression in perichondrial cells correlates with the absence of both growth plates and longitudinal growth in the stylopod and zeugopod elements; a phenotype consistent with the necessity for Wnt5a to promote hypertrophic chondrocyte differentiation, growth plate formation and proliferation in the primary ossification center[36,37]. Thus, our findings suggest that expression of Creb5 in the articular perichondrium may both direct this tissue to give rise to articular cartilage (in a cell autonomous fashion); and by repressing the expression of Wnt5a in the articular perichondrium may block the formation of growth plate formation in the adjacent epiphyseal chondrocytes, which thus allows the formation of secondary ossification centers in the epiphyses.

While our work indicates that forced expression of iCreb5-HA<sup>(Prx1-Cre)</sup> in the perichondrium represses that of Wnt5a and Crabp1 in this tissue, it is not clear whether the repression of Wnt5a/Crabp1 expression is either a direct effect of Creb5 or an indirect effect of one or more of the downstream genes induced by Creb5 in the articular perichondrium (diagrammed in Fig. 10d). It is possible that Creb5-dependent exclusion of Wnt5a expression from both the developing joint interzone and articular perichondrium may establish a region of competence for the "canonical" Wnt signaling pathway to promote joint formation, as this signaling pathway is inhibited by Wnt5a[64–66]. Indeed, while Wnt5a expression encroached into the perichondrium adjacent to the fused tibiofemoral cartilage element in Creb5<sup>Δ9/Δ9</sup> mice; the few synovial joints that still formed in these animals (e.g., the ulnohumeral and mesopodial joints) continued to lack Wnt5a expression in their interzones (Supplementary Fig. 10),

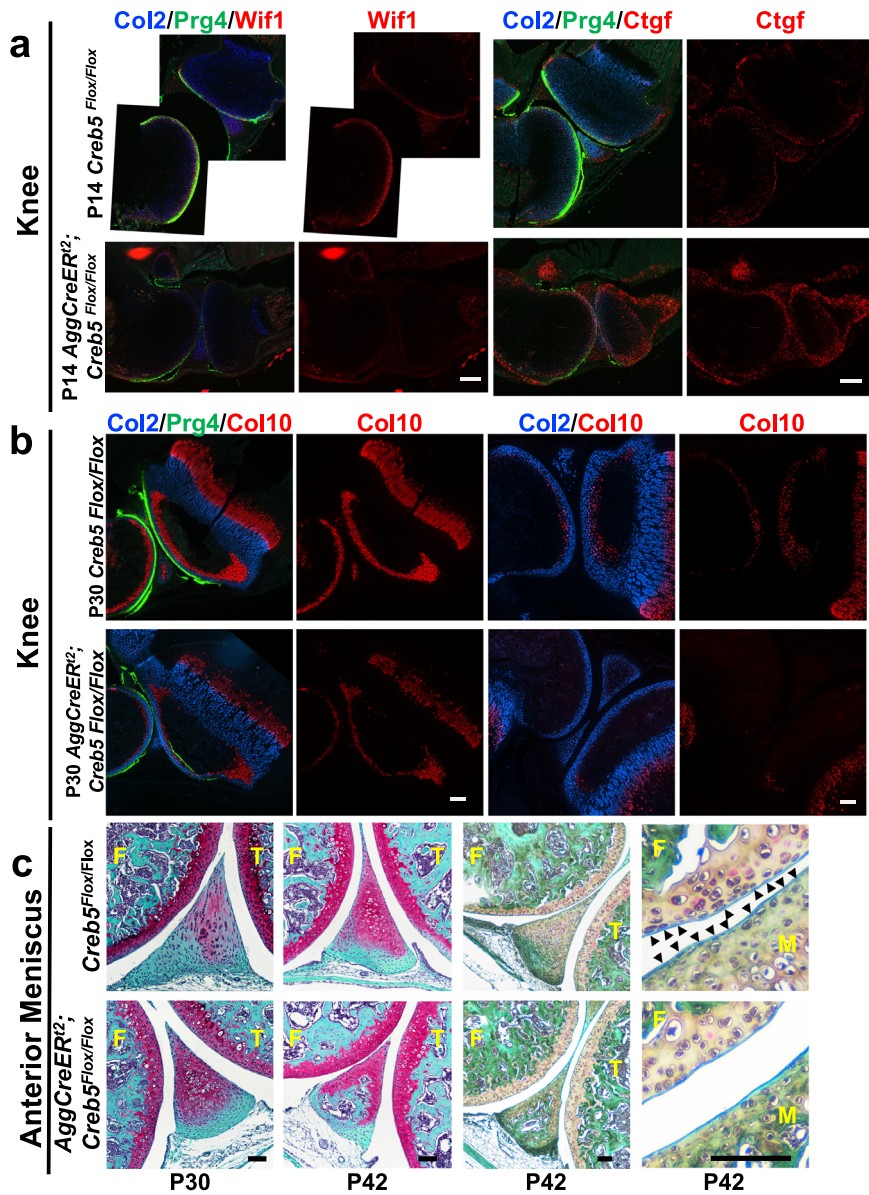

**Fig. 9 | Creb5 is necessary to maintain appropriate zonal gene expression in the articular cartilage and blocks premature endochondral ossification of adjacent tissues in postnatal mice.** Gene expression assayed by RNAscope FISH in the hindlimbs of either P14 (**a**) or P30 (**b**) *Aggrecan1^tm(IRES-CreERT2)*; *Creb5^flox9/flox9* mice or *Creb5^flox9/flox9* littermates, that had been administered tamoxifen from postnatal days 1–11. In each row of images, the left two images depict the same section; and the right two images depict the same section. Scale bar equals 200 microns in **a** and **b**. **c** Safranin O/Fast Green staining of the anterior meniscus (and articular cartilage) in the knees of either P30 or P42 *Aggrecan1^tm(IRES-CreERT2)*; *Creb5^flox9/flox9* mice or *Creb5^flox9/flox9* littermates, that had been administered tamoxifen from postnatal days 1–11. Arrowheads indicate flat cells on the surface of the meniscus. Femur (F), Tibia (T), and Meniscus (M) are indicated. Scale bar equals 100 microns in **c**. P14 images were obtained from tissues of female mice; P30 and P42 images were obtained from tissues of either male or female mice. Similar results were obtained in at least 2 (**a**, **b**) or 3 (**c**) independent biological repeats.

despite the absence of Creb5 function. Thus, the synovial joints which form in the absence of Creb5 do not require this transcription factor to repress Wnt5a expression in either their interzone or articular perichondrium. Taken together, our findings suggest that Creb5 maintains the competence for synovial joint formation by promoting the expression of both Wnt family members that stabilize β-catenin and Gdf5, which attenuates BMP signaling by BMPR1a[48] (reviewed in ref. [49]); while simultaneously blocking the expression of Wnt5a in the developing joint interzone.

We have found that maintained expression of Creb5 in postnatal articular cartilage is necessary to sustain appropriate zonal expression of both signaling molecules and Prg4 expression in this tissue; and is thus crucial to maintain the health of the synovial joint. In addition to decreased expression of Prg4 and Wif1 in articular cartilage, postnatal

deletion of Creb5 in aggrecan-expressing cells resulted in a decreased number of Col10a1-expressing cells on the side of the secondary ossification center that is nearest the articular surface, premature endochondral ossification of cells in the anterior meniscus, and a loss of flat cells in the superficial zone of the articular cartilage. Thus, postnatal loss of Creb5 (in articular cartilage) affects both Creb5-expressing cells as well as tissues that lie adjacent to Creb5-expressing cells. These findings are consistent with our identification of Creb5 transcriptional target genes in bovine articular chondrocytes to include Wif1, Gdf10, Fat1, Fat4, EphA3, Hbegf (an EGFR-ligand) and Tgfbr1; suggesting that Creb5 coordinates several distinct signaling pathways to maintain the health of the articular cartilage. By inducing the expression of Wif1, we speculate that Creb5 may indirectly modulate the intensity of Wnt signaling in the synovial joint and the

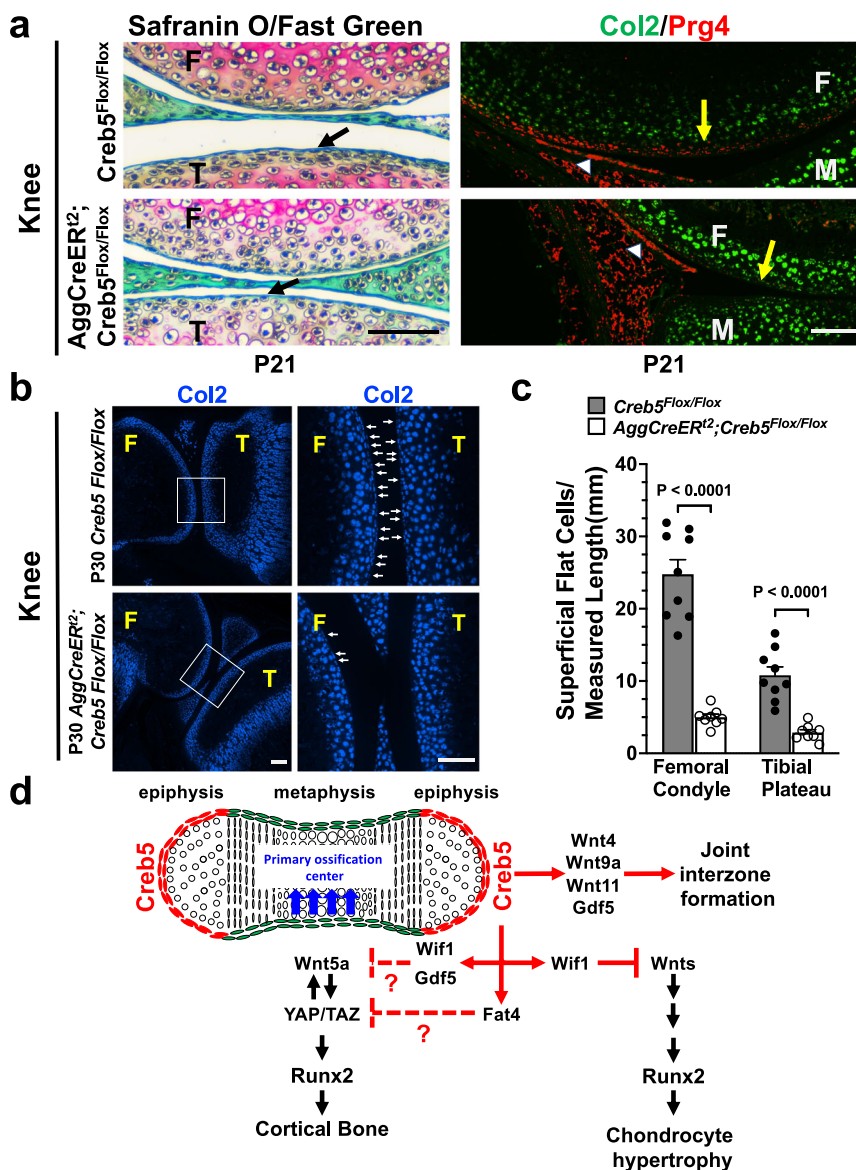

**Fig. 10 | Creb5 is necessary to maintain flat cells in the superficial zone of the articular cartilage in postnatal mice. a** Safranin O/Fast Green staining or RNA-Scope FISH of the articular surface of the knees of P21 *Aggrecan1^tm(IRES-CreERT2)*; *Creb5^flox9/flox9* mice or *Creb5^flox9/flox9* littermates, that had been administered tamoxifen from postnatal days 1–11. Left: Location of normally flat cells (in control mice) in the superficial zone of the tibial articular cartilage is indicated (black arrows). Right: Note that in *Aggrecan1^tm(IRES-CreERT2)*; *Creb5^flox9/flox9* mice, Aggrecan-CreERt2 mediated deletion of exon 9 from *Creb5^flox9/flox9* reduces Prg4 expression in the surface of the articular cartilage (yellow arrows) but does not affect Prg4 expression in the synovial membrane (which does not express Aggrecan-CreERt2) (white arrowheads). Femur (F), Tibia (T) and Meniscus (M) are indicated. Scale bar equals 100 microns in **a**. P21 images were obtained from tissues of female mice. **b** RNAScope FISH of Col2a1-expressing cells in the knee joint of either a P30 male *Aggrecan1^tm(IRES-CreERT2)*; *Creb5^flox9/flox9* mouse or a P30 male *Creb5^flox9/flox9* littermate, that had been administered tamoxifen from postnatal days 1–11. Box outlined in left image is enlarged in right image. Arrows indicate small flat Col2a1-expressing cells on the surface of the articular cartilage. Femur (F) and Tibia (T) are indicated. Scale bar equals 200 microns in left panel; 100 microns in right panel. Similar results were obtained in at least 4 (**a**, **b**) independent biological repeats. **c** Quantitation of the number of small flat cells/mm measured surface length of the articular cartilage displayed in **b**. Each point represents a different RNAScope section obtained from either one P30 male *Creb5^flox9/flox9* mouse that was treated with tamoxifen ($n = 9$ sections); or from one P30 male *Aggrecan1^tm(IRES-CreERT2)*; *Creb5^flox9/flox9* mouse that was treated with tamoxifin ($n = 8$ sections). Statistical analysis was a two-sided test without adjustments. *p* values are indicated and error bar indicates standard error of the mean. **d** Creb5 is critical to maintain appropriate expression of both Wnt ligands and other signaling molecules in the interzones of developing synovial joints; and is thus crucial for the formation of most joints. Creb5 both promotes the formation of articular chondrocytes and blocks growth plate development in the epiphyses by inhibiting expression of Wnt5a in the articular/epiphyseal perichondrium.

consequent endochondral ossification of periarticular tissues, which is driven by Wnt signaling (reviewed in ref. 67).

## Methods

### Vertebrate animals
All work with vertebrate animals was approved by the Harvard Medical School Institutional Animal Care and Use Committee (IACUC).

### Isolation and culture of bovine articular chondrocytes
The knee joints from 1–2 week old bovine calves were obtained from Research 87 (a local abattoir in Boylston, MA) directly after slaughter. The intact femoropatellar joint was isolated by transecting the femur and mounting the distal segment. The femoropatellar articular cartilage was then exposed by opening the joint capsule, severing the medial, lateral, and cruciate ligaments, and removing the tibia,

patella, and surrounding tissue. During this entire process, the cartilage was kept moist and free of blood by frequent rinsing with sterile PBS supplemented with antibiotics (100 U/ml penicillin and 100 μg/ml streptomycin). An initial ~200–300 micron thick slice of superficial zone articular cartilage (SZC) was first harvested by scalpel. The next ~4 mm thick slice of middle zone cartilage tissue was then removed by scalpel and discarded. Finally, an 800 micron to 1 mm thick slice of deep zone articular cartilage (DZC) was then harvested by scalpel. All of the superficial zone slices from a given knee joint were placed in a 50 mL centrifuge tube filled with medium (DMEM with 10 mM HEPES, 0.1 mM nonessential amino acids, and additional 0.4 mM proline, 25 pg/ml ascorbate) and supplemented with 10% fetal bovine serum. Similarly, all deep zone slices from a given joint were placed in a separate tube filled with medium and serum. To isolate chondrocytes from the superficial and deep zone slices, deep zone cartilage shavings were chopped into pieces about 1 mm$^3$. There was no need to chop superficial zone shavings as they are thin enough. Both superficial zone and chopped deep zone cartilage were digested in 10 ml of pronase (1 mg/ml) in DMEM with 1% penicillin & streptomycin for 1 h. Pronase was replaced by collagenase D (1 mg/ml) in DMEM with 1% penicillin & streptomycin and cartilage tissue was digested in the incubator at 37 degrees overnight. The next day, the dissociated cells were filtered through a 70 μm strainer, counted, pelleted for 5 min at 250 g and resuspended in DMEM/F12 plus 10% FBS. For lentivirus infection the cells were plated into a 6-well plate (2–3 million cells per well). 24 h after plating the medium was changed with new DMEM/F12 plus 10% FBS.

### Construction of lentivirus encoding WT or mutant Creb5

Total RNA (containing Creb5 transcripts) was isolated from bovine superficial zone articular chondrocytes using Trizol reagent (Life Technologies). cDNA was reverse transcribed using the oligo dT reverse transcription kit SuperScript® III First-Strand Synthesis System (Life Technologies, cat. no. 18080051). The bovine Creb5 open reading frame (508aa; see NM_001319882.1) was predicted by RNA-Seq of bovine superficial zone articular chondrocytes[18]. A 1524 bp cDNA fragment (encoding 508aa) was PCR amplified from bovine superficial zone articular chondrocyte cDNA using the Q5 High-Fidelity 2X Master Mix (NEB, cat. no. M0492S) and then cloned into the pCR®-Blunt vector (Life Technologies, Cat#: K270020). Using the pCR®-Blunt-bovine Creb5 as a template, a fragment of DNA encoding 3xHA tags was added onto the C-terminus of Creb5 (immediately before the stop codon). HA-tagged Creb5 was then cloned into a Gateway vector (pENTR-Creb5-HA) using the pENTR™/SD/D-TOPO® Cloning Kit (Life Technologies, Cat#: K2420-20). Creb5-HA was then transferred from pENTR-Creb5-HA into the pInducer20 (Addgene # 44012) lentivirus destination vector using Gateway LR Clonase (Thermo Fisher, cat. no. 11791020), to generate pInducer20-iCreb-HA. To generate lentivirus vectors encoding Creb5(Δ9)-HA, the DNA sequence corresponding to the 9th exon of *Creb5* DNA was deleted from pENTR-Creb5-HA using the Q5® Site-Directed Mutagenesis Kit (NEB, Cat#: E0554S) following the manufacturer's instructions. The sequence of either Creb5(WT)-HA or Creb5(Δ9)-HA (cloned into pENTR) were confirmed by sequencing; and both then cloned into the lentivirus destination vector pInducer20 using Gateway technology.

### Growth and purification of lentivirus

HEK293 cells were used to package lentivirus. HEK293 cells were plated in a 15-cm dish in 25 ml of DMEM/F12 (Invitrogen) supplemented with 10% heat-inactivated fetal bovine serum (Invitrogen) and Pen/Strep at 37 °C with 5% CO$_2$. Transfection was performed when the cells were ~70–80% confluent. Lentiviral expression plasmid (6 μg), psPAX2 (4.5 μg, Addgene #2621), and pMD2.G VSVG (1.5 μg, Addgene #2622) plasmids were added into a sterile tube containing 500 μl of Opti-MEM® I (Invitrogen). In a separate tube, 36 μl of Fugene 6 was diluted

into 500 μl of Opti-MEM I. The diluted Fugene 6 reagent was added drop-wise to the tube containing the DNA solution. The mixture was incubated for 15–25 min at room temperature to allow the DNA-Fugene 6 complex to form. The DNA-Fugene 6 complex was directly added to each tissue culture dish of HEK293 cells. After cells were cultured in a CO2 incubator at 37 °C for 12–24 h, the medium (containing the DNA-Fugene 6 complex) was replaced with 36 ml fresh DMEM/F12 medium supplemented with 10% heat-inactivated fetal bovine serum and penicillin-streptomycin. Cells were again placed in the CO2 incubator at 37 °C; and virus-containing culture medium was collected in sterile capped tubes at 48, 72 and 96 h post-transfection. Cell debris was removed by centrifugation at 500 × *g* for 10 min, and filtration through nylon low protein-binding filters (SLHP033RS, Millipore). Virus was concentrated by ultracentrifugation (employing a SW32 rotor at 106883 g for 2 h and 30 min at 4 °C) and stored at −80° C.

### Infection with lentivirus and RT-qPCR analysis

Newly isolated bovine superficial zone articular chondrocytes or deep zone articular chondrocytes were cultured for at least 2–3 days before infection. Ultracentrifuge-concentrated lentivirus was added into DMEM/F12 medium (with 10% FBS) containing 7.5 μg/ml DEAE-Dextran (to increase infection efficiency). In total, 24 h after infection, medium was replaced with new DMEM/F12 medium (with 10% FBS) containing either 0.8 μg/ml puromycin or 500 μg/ml G418, for selection. After selection (for either 5 days in puromycin or 11 days in G418) the cells were re-plated onto low attachment tissue culture plates (Corning #3471 or #3473) in DMEM/F12 medium (with 10% FBS), without either puromycin or G418. Chondrocytes were cultured in the presence of doxycycline, TGF-α and TGF-β2 as indicated in Supplementary Table 4. After 2–3 days culture, RNA was harvested using Trizol reagent and cDNA was synthesized using SuperScript™ III First-Strand Synthesis SuperMix (Invitrogen, Cat#: 11752-050) according to the manufacture's guidelines. RT-qPCR primers were synthesized by Integrated DNA Technologies. RT-qPCR was performed in an Applied Biosystem 7500 Fast Real-Time PCR Machine. Each experiment was performed with 2–3 biological repeats and each RT-qPCR assay was performed with two technical repeats. Prism statistical software (GraphPad) was employed to analyze the data. Statistical analysis was performed using paired *t* tests. In RT-qPCR experiments, gene expression was normalized to that of *GAPDH*, as indicated. All RT-qPCR primers are listed in Supplementary Table 1.

### Chromatin IP (ChIP)-PCR analysis

Bovine deep zone articular chondrocytes infected with lentivirus encoding either doxycycline-inducible HA-tagged Creb5 (iCreb5-HA) or doxycycline-inducible HA-tagged Creb5(Δ9) (iCreb5(Δ9)-HA). The cells were cultured in ultra-low attachment tissue culture dishes (Corning, Cat#: 3471) for 2 days in DMEM/F12 supplemented with 10% FBS and TGF-α plus TGF-β2, in either the absence or presence of doxycycline (1 μg/ml). The chondrocytes were gently centrifuged (200 × *g*, 5 min), and digested with 0.25% Trypsin-EDTA (Thermo Fisher, Cat#: 25200056) at 37 °C for 7 min (until the tissue was dispersed into single cells). Trypsinization was stopped by adding 10x volume 10% FBS/DMEM. Digested single cells were fixed in 1% formaldehyde at room temperature for 20 min with shaking. Formaldehyde was quenched with 125 mM glycine on a shaking platform at room temperature for 10 min. Cells were then washed two times with cold PBS containing protease inhibitor cocktail (Roche) followed by lysis with 500 μl ChIP sonication buffer (0.5% SDS, 10 mM EDTA, 50 mM Tris-HCl pH 8.1). Chromatin (from ~5 million cells in 500 μl ChIP sonication buffer) was sonicated with a Covaris E220 Sonication Machine (PIP = 140, CBP = 200, DF = 5%, Avg Power = 7.0, 300 s each time, 6 times (Total 30 min). Debris was removed by spinning samples 8 min at 20,800 g at 4 °C. The chromatin was diluted with ChIP Dilution Buffer (1.1% TritonX-100, 1.2 mM EDTA,

16.7 mM Tris-HCL pH 8.1, 167 mM NaCl) up to 1500 ul in 1.5 ml eppendorf tube (DNA low bind tube). SDS concentration was adjusted to below 0.2% and TritonX-100 concentration was adjusted to 1%. Five micrograms of antibody (against the antigen of choice) was added to the chromatin and immunoprecipitated overnight at 4 °C. Protein A (15 µl) and Protein G (15 µl) beads were washed two times with ChIP Dilution Buffer and then added to the chromatin. The bead-chromatin complex was rotated at 4 °C for 2 h and then washed two times each with Low Salt Buffer (0.1% SDS, 1% TritonX-100, 2 mM EDTA, 20 mM Tris-HCl pH 8.1, 150 mM NaCl), High Salt Buffer (0.1% SDS, 1% TritonX-100, 2 mM EDTA, 20 mM Tris-HCl pH 8.1, 500 mM NaCl), Lithium Buffer (0.25 M LiCl, 1% IGEPAL CA630 (or NP-40), 1% Deoxycholic acid (Sodium Salt), 1 mM EDTA, 10 mM Tris pH8.1), and finally TE Buffer (10 mM Tris-HCl pH8.0, 1 mM EDTA). Chromatin-antibody complexes were eluted twice from beads using 100 µl of Elution Buffer (1% SDS, 0.1 M NaHCO3) by incubation at room temperature for 10 min. NaCl was added into the 200 µl elute to a final 370 mM concentration. Cross-linked chromatin was reversed at 65 °C with shaking overnight followed by adding RNase A (to a final concentration of 0.2 mg/ml) and incubating at 37 °C for 1 h to digest RNA; and then adding Proteinase K (to a final concentration of 0.1 mg/ml) and incubating at 55 °C for 1 h to digest protein. DNA fragments were purified using the PCR-purification kit with Min-Elute Columns (Qiagen) and analyzed using qPCR. For ChIP-qPCR quantitation, the level of immunoprecipitated DNA was normalized to that of the input DNA. Antibody employed for ChIP was rabbit anti-HA (Cell Signaling Technology; Cat# 3724S, 15 ul per ChIP). Primers employed to amplify the E1 enhancer of the bovine *Prg4* locus[18] were Forward: aaaaacactctttgctgg; Reverse: agatggtaaattgcttggg.

### RNA-Seq following shRNA-knockdown of Creb5

Newly isolated bovine superficial zone articular chondrocytes were cultured (in DMEM/F12 plus 10% FBS) for 2 days before infection. Ultracentrifuge-concentrated shScramble (Addgene Plasmid #1864) or shCreb5 lentivirus[18] was added into DMEM/F12 medium (with 10% FBS) containing 7.5 µg/ml DEAE-Dextran (to increase infection efficiency). Twenty-four hours after infection, medium was replaced with new DMEM/F12 medium (with 10% FBS) containing 0.8 µg/ml puromycin for selection. After puromycin selection for 5 days, shScramble or shCreb5 infected superficial zone articular chondrocytes were replated onto low attachment tissue culture plates (Corning #3471 or #3473) in DMEM/F12 medium (with 10% FBS) in either the presence of TGF-α (100 ng/ml) alone (condition A) or in the presence of both TGF-α (100 ng/ml) and TGF-β2 (20 ng/ml) (condition AB). After 2 days of culture in low attachment tissue culture plates, RNA from conditions A and AB were collected using Trizol reagent (Life Technologies). RNA was isolated from lentivirus infected superficial zone articular chondrocytes (that had been cultured under either condition A or condition AB) from 4 independently collected bovine knees. mRNA was purified and used to prepare the libraries using the KAPA mRNA HyperPrep Kits (Roche) following the manufacturer's instructions. RNA quality and quantity were assessed on an Agilent TapeStation. In all 16 RNA-Seq samples (isolated from 4 different animals), 14 of 16 samples had an RNA integrity number in the 8.2–9.1 range, only 2 of 16 samples had an RNA integrity number in the 7–8 range (i.e., 7.1 and 7.7.). For RNA-Seq library preparations, mRNA was isolated using oligo-dT beads and the resulting mRNA was converted into cDNA. The cDNA libraries were prepared by adapter ligation and post-PCR cleanup. Single-end sequencing was performed using NextSeq 500 High-Output 75-cycle kit and NextSeq 500 Mid-Output 75-cycle kit on a NextSeq 500 System.

### Bioinformatics analysis of RNA-Seq libraries

The sequencing data were processed using the bcbio-nextgen pipeline (bcbio/bcbio-nextgen: v1.2.4 (v1.2.4). Zenodo. https://doi.org/10.5281/zenodo.4041990) with bovine reference genome UMD 3.1. The basic quality of the data was checked using FastQC[68] and mapping metrics were assessed by aligning the reads to reference genome using STAR[69]. Transcript expression was quantified using Salmon[70] and aggregated for gene-level counts using tximport[71]. Differentially expressed (DE) genes between treatment pairs were identified using DESeq2[72] with FDR threshold of 0.05. Sample IDs of the bovine calves were included as a covariate in the linear model for DESeq2.

### Statistics and reproducibility

RNA-Seq was performed with quadruplet biological samples. RT-qPCR assays were performed with 2–3 independent biological repeats and each assay was performed with two technical repeats. Histological analyses, Fluorescent In-Situ Hybridization (FISH) and RNAScope FISH were each performed with tissues from at least 2-3 embryos/mice (with multiple serial sections of all tissues analyzed) for any given genotype and/or treatment. Representative images are displayed. GraphPad Prism 9 was employed for statistical analyses using two-tailed Student's *t* test.

### Histological techniques

Whole mount Alcian Blue/Alizarin Red staining: after euthanization (by $CO_2$), the skin of P0 mice was carefully removed. All the organs in the peritoneal and pleural cavities were removed with forceps. The tissue was fixed in 95% ethanol overnight at room temperature on a gentle rocker, followed by another overnight acetone incubation to remove fat. Tissue was washed with deionized water and then stained with Alcian Blue (0.3%, Sigma A5268)/Alizarin Red (0.1%, Sigma 5533) solution at 37 °C for 1 day; followed by incubation at room temperature for another 2 days. After the staining, the tissue was rinsed with deionized water and then put into 1% KOH. Soft tissue was removed by 2–3 days incubation in 1% KOH. The skeletons were then sequentially placed for at least 24 h (at room temperature) in each of the following solutions: 20% glycerol containing 1% KOH; 50% glycerol containing 1% KOH; 80% glycerol containing 1% KOH, and stored in 80% glycerol at room temperature.

Safranin O/Fast Green: forelimbs and hindlimbs were dissected and fixed in 4% paraformaldehyde (in PBS) overnight at 4° C, and decalcified in 14% EDTA overnight at 4° C. The tissues were subsequently washed with PBS on ice for 10 min, and then incubated at room temperature for 10 min in each of the following solutions: ETOH (25%) in PBS; ETOH (50%) in PBS; ETOH (75%) in PBS; 100% ETOH. After dehydration, the tissues were placed into the following solutions: 100% xylene (30 min at room temperature); 100% xylene (30 min at room temperature); 50% xylene/50% paraffin (15 min at 60° C); 100% paraffin (overnight at 60° C). Tissues were embedded in paraffin and sections were cut at 6 µm thickness using a microtome. Safranin O/Fast Green staining was performed following the protocol from the University of Rochester Medical Center (https://www.urmc.rochester.edu/musculoskeletal-research/core-services/histology/protocols.aspx).

Von Kossa staining: Von Kossa staining was performed on undecalcified cryosections using a silver plating kit (EMD Millipore, Cat. 100362) per manufacturer's instructions. Forelimbs or hindlimbs were dissected and fixed in 4% paraformaldehyde at 4 °C overnight. Tissues were embedded in OCT and frozen sections were cut at 20 µm using a cryostat. After Von Kossa staining, a counterstain was performed with nuclear fast red solution followed by Alcian Blue solution.

### Western blots

Bovine deep zone articular chondrocytes or mouse limbs were collected and lysed in Lysis Buffer (containing 50 mM Tris· HCl, pH 7.4, 1% NP-40, 0.5% Sodium deoxycholate, 1% SDS, 150 mM NaCl, 2 mM EDTA, 50 mM NaF, protease inhibitor cocktail (Roche)). The cell lysate were incubated on ice for 15 min, and centrifuged at $3000 \times g$, at 4 °C to remove cell debris. Protein concentration was measured with a Bio-Rad Protein Assay Kit (BIO-RAD, Cat#: 500-0006). The protein was

denatured at 70 °C for 10 min with LDS Sample Buffer (Invitrogen, Cat#: NP0007) and Sample Reducing Agent (Invitrogen, Cat#: NP0009). General SDS-PAGE processing procedures were followed. The blots were visualized by the enhanced chemiluminescence detection method, employing the Pierce ECL Western Blotting Substrate (Pierce, Cat#: 32106). All primary antibodies used for Western blots are listed in Supplementary Table 2.

### Immunohistochemical analysis

Mouse limbs were dissected and fixed in 4% paraformaldehyde (in PBS) at 4 °C overnight, washed with PBS, and incubated in 30% sucrose at 4 °C overnight. Tissues were embedded in OCT and frozen sections were cut at 12 μm using a cryostat. Any non-specific binding of primary antibodies was blocked by incubation with PBS containing 0.2% Tween-20 and 5% non-immune goat serum for 1 h at room temperature. Primary antibody incubation was carried out at 4 °C overnight with a rabbit polyclonal antibody to Creb5 (PA5-65593, Thermo Fisher, 1:100), a rat monoclonal antibody to the HA epitope (11867423001, Sigma, 1:100), a rabbit antibody to the HA epitope (3724S, Cell Signaling Technology, 1:200), a rabbit antibody to phospho-Smad2 (Ser465/Ser467) (18338S, Cell Signaling Technology, 1:50) or a rabbit polyclonal antibody to Sox9 (AB5535, EMD Millipore, 1:100). After washing in PBST (3 × 15 min at room temperature), sections were incubated with Alexa Fluor 488 or 594 conjugated secondary antibodies (Thermo Fisher, 1:250) for 1 h at room temperature. 4,6-diamidino-2-phenylindole (DAPI; 1 μg/ml) was included with the secondary antibodies to stain DNA. Slides were washed in PBST (3 × 15 min at room temperature) and mounted under a coverslip with Aqua-mount (13800; Lerner Labs). Images were taken at the Nikon Imaging Center at Harvard Medical School. All antibodies employed for immunohistochemistry are listed in Supplementary Table 3.

### In situ hybridization analysis

Digoxigenin (DIG) or fluorescein labeled RNA probes were made using the DIG RNA labeling kit (Roche, Cat. 11175025910) or fluorescein RNA labeling kit (Roche, cat. no. 11685619910), respectively, per manufacturer's protocol. T7 RNA polymerase (Roche, cat. no. 10881767001), T3 RNA polymerase (Roche, cat. no. 11031171001) or SP6 RNA polymerase (Roche, cat. no. 10810274001) were employed to generate RNA, depending on the particular RNA probe. Target sequence (sense strand) of in situ hybridization probes listed in Supplementary Table 5. After DNase I treatment and precipitation, RNA probes were dissolved in 40 μl RNAse-free water. Forelimbs or hindlimbs were dissected and fixed in 4% paraformaldehyde (made with DEPC-treated PBS) at 4 °C overnight, washed with PBS, and incubated in 30% sucrose (made with DEPC-treated H2O) at 4 °C overnight. Tissues were embedded in OCT and frozen sections were cut at 20 μm using a cryostat. Sections were fixed again with 4% paraformaldehyde (made with DEPC-treated PBS) at room temperature for 5 min, washed twice (5 min each at room temperature) with PBS containing 0.1% Tween 20 (PBST). The sections were digested with Proteinase K (4 μg/ml in PBS) at room temperature for 8–20 min (depending on the RNA probe), washed twice with PBST (5 min each) and fixed again with 4% paraformaldehyde (in PBS) at room temperature for 5 min. After washing twice with PBST (5 min each, at room temperature), sections were acetylated for 10 min (at room temperature) in a solution containing 0.25% acetic anhydride and 0.1 M triethanolamine (in DEPC-treated H2O). After washing twice with PBST (5 min each, at room temperature) the slides were rinsed in DEPC-treated H2O, and then air dried. The RNA probe was diluted to 100 ng/ml in a 200 μl volume of hybridization solution (10 mM Tris–HCl (pH 7.5), 600 mM NaCl, 1 mM EDTA, 0.25 % SDS, 10 % dextran sulfate, 1x Denhardt's solution, 200 μg/ml yeast tRNA, 50% formamide), and heated at 95 °C for 5 min before adding to the slide. Hybridization was performed at either 65 °C or 70 °C overnight, and washed according to the protocol outlined in ref. 73. After

hybridization, washes, and blocking endogenous peroxidase activity, the slides were blocked with 10% heat-inactivated goat serum at room temperature for 1 h. Anti-DIG-POD antibody (1:100 dilute, Anti-Digoxigenin-POD, Fab fragment, 11207733910, Roche) or Anti-Fluorescein-POD antibody (1:100, Anti-Fluorescein-POD, Fab fragments, 11426346910, Roche) was added and the slides were incubated at 4 °C overnight. After washing in TNT Buffer (100 mM Tris–HCl pH 7.5, 150 mM NaCl, 0.05% Tween20), the biotin (1:100) amplification reagent (TSA Biotin Kit, NEL749A001KT) or Fluorescein (1:100) amplification reagent (TSA plus Cyanine 3/ Fluorescein System, NEL753001KT) was applied and slides incubated 20 min at room temperature to develop the signal. Biotin signal was further detected by Streptavidin Secondary Alexa 594 dyes. Slides were washed in TNT Buffer (3 × 10 min at room temperature), rinsed in H2O, and mounted under a coverslip with Aqua-mount (13800; Lerner Labs).

### Multiplex RNAscope FISH

For RNAscope (Advanced Cell Diagnostics) analyses, tissues were first fixed in 4% paraformaldehyde (made with DEPC-treated PBS) at 4 °C overnight, washed with PBS, and incubated in 30% sucrose (made with DEPC-treated H2O) at 4 °C overnight, then embedded in OCT and frozen sections were cut at 20 μm using a cryostat as described above. All the protocols were performed exactly following the manufacturer's instructions (RNAscope® Multiplex Fluorescent V2 Assay). The following probes were used: Mm-Col2a1 (Cat# 407221-C4), Mm-Creb5 (Cat# 572891-C3), Mm-Gdf5 (Cat# 407211), Mm-Wnt5a (Cat# 316791-C2), Mm-Wif1 (Cat# 412361-C1), Mm-Ctgf (Cat# 314541-C1), Mm-Wnt4 (Cat# 401101-C1), Mm-Wnt9a (Cat# 405081-C1), Mm-Wnt11 (Cat# 405021-C1), Mm-Fat4 (Cat# 447511-C1), Mm-Col10a1 (Cat# 426181-C1), Mm-Prg4 (Cat# 437661-C3), Mm-Crabp1(Cat# 474711), and Mm-Msx1 (Cat# 421841-C2).

### Quantitation of flat cells in the articular cartilage

We mated *Creb5^flox9/flox9* mice with *Aggrecan1^tm(IRES-CreERT2);Creb5^flox9/flox9* mice to generate litters containing both *Aggrecan1^tm(IRES-CreERT2);Creb5^flox9/flox9* mice or *Creb5^flox9/flox9* control mice (which lack the CreERt2 driver allele). Tamoxifen was repeatedly administered to these litters at postnatal days 1–11, to specifically delete exon 9 of *Creb5^flox9/flox9* in postnatal stage articular chondrocytes. Following sacrifice at P30, the knees of these animals were fixed, processed, paraffin-embedded, and sectioned. The location and morphology of Col2a1-expressing cells in the articular cartilage was assayed by RNAScope analysis. To quantitate the number of flat cells in the superficial zone of the articular cartilage, we counted (in a double-blinded fashion) the number of flat cells present in the superficial most top two layers of cells, normalized to the length of the measured articular cartilage surface.

### Generation of *Creb5^Δ9/+* mice

We employed CRISPR/Cas9 technology[74,75] to generate the *Creb5^Δ9* allele, which encodes a mutant form of Creb5 that lacks an intact basic-leucine zipper DNA binding domain of this protein. Guide RNAs (gRNAs) to guide Cas9 to introns that flank the 9th exon of *Creb5* (~200 nucleotides upstream and downstream of the 9th exon) were designed following the instructions at (http://crispr.mit.edu/). To identify gRNAs that efficiently directed Cas9 cutting within the *Creb5* gene, we designed three gRNAs to sequences in each intron that flanks the 9th exon of *Creb5*. gRNAs were cloned into px459 (Addgene # 62988) and transfected into NIH3T3 cells. After puromycin selection of transfected NIH3T3 cells, the T7 endonuclease I (T7E1) assay was performed to test the gRNA cutting efficiency. The gRNAs with the highest cutting efficiency in each flanking intron were chosen and synthesized (Synthego, CAUUUGGAGAAUCCUAGUGG—Modified, and UUGGUCCCUAAUUAU AGAUG—Modified). Cas9 protein was purchased from PBA Bio (cat. no. CP01-50). Guide RNAs plus Cas9 protein were injected into C57BL/6

one cell mice embryos[74,75] by the Genome Modification Facility at Harvard University. Of 33 pups that were born, four animals contained an allele of *Creb5* that had undergone non-homologous end-joining following Cas9 cutting, and thus had deleted the 9th exon. One (mouse #8) of these four animals was mated to C57/BL6 mice to generate the *Creb5^Δ9/+* mouse line. The following primers were used for genotyping, F3: CCCATTTGCAGTCTGTAG; R3: GATGTTCC-TAGCTAAGCC. The 700 bp PCR amplicon (employing these primers) was generated from founder mouse #8 genomic DNA. Sequence analysis of this amplicon derived from genomic DNA revealed that exon 9 was deleted in the *Creb5^Δ9* allele (Supplementary Fig. 1a, b). Consistent with loss of exon 9 in the *Creb5^Δ9* allele, sequencing of the Creb5 cDNA (derived from brain tissue of either adult *WT* or adult *Creb5^Δ9/+* mice) with primers located in *Creb5* exon 8 (Exon 8F:ACATGATGGAGATGATGGGCTC) and in *Creb5* exon 11 (Exon 11R:GTGTTGGATAACTTGCTGCTGA) indicated that exon 8 is directly spliced (in frame) to exon 10 in the *Creb5^Δ9* mRNA (Supplementary Fig. 1c, d). Deletion of exon 9 is predicted to yield an in-frame mutant Creb5 protein, that specifically lacks 76 amino acids that encode a critical part of the bZIP domain of Creb5 and is necessary for direct interaction of Creb5 with DNA targets, such as the E1 enhancer that drives Prg4 expression (Supplementary Fig. 1f).

### Generation of *Creb5^flox9/flox9* mice

We employed CRISPR/Cas9 technology plus a long single strand DNA donor as a recombination template to generate the *Creb5^flox9/+* mouse line. The same gRNAs with the highest cutting efficiency in the flanking introns of exon 9 of *Creb5* (employed to generate the *Creb5^Δ9/+* mouse line) were also employed to generate the *Creb5^flox9/+* mouse line (Synthego, CAUUUGGAGAAUCCUAGUGG−Modified, and UUGGUCC-CUAAUUAUAGAUG−Modified). A 1369-base single strand DNA donor containing a floxed version of exon 9 of *Creb5*, flanked leftward (by 270 base) and rightward (by 270 base) homologous arms, was synthesized by Integrated DNA Technologies. Guide RNAs, the 1369-base single strand DNA donor, and Cas9 protein (PBA Bio, Cat# CP01-50) were injected into C57BL/6 one cell mice embryos[74,75] by the Genome Modification Facility at Harvard University. The same primers used for genotyping the *Creb5^Δ9/+* mice were also used to identify the *Creb5^flox9/+* mice; F3: CCCATTTGCAGTCTGTAG; R3: GATGTTCCTAGCTAAGCC. Both F3 and R3 primers lie outside the left and right homologous arms that were present in the 1369- base single strand DNA donor. F3 and R3 amplify a 1420 bp amplicon from the wildtype *Creb5* genomic locus. In contrast, after introduction of LoxP sites flanking exon 9 of *Creb5*, F3 and R3 primers amplify a 1500 bp amplicon containing left and right homologous arms, two loxP sites and exon 9 of *Creb5*. The 1500 bp genomic fragment amplified with F3 and R3 primers from founder mouse #3 genomic DNA (Supplementary Fig. 9a, b) was ligated into a PCR-Blunt vector (Invitrogen), and the whole fragment was sequenced by Genewiz (Supplementary Table 6). Germline transmission was achieved by crossing the #3 mouse with a C57BL/6 mouse. The following primers were used for subsequent genotyping of the left loxP site and the right loxP site, F1: AAGGCTCTTCTGGAGGAG; R1: CCTGCAGGACATTTGGAG; and F2: CTGGCTCTCTGTTTCCTC; R2: GT TCCTAGCTAAGCCACC.

### Conditional deletion of Creb5 in postnatal articular cartilage

An *Aggrecan1^tm(IRES-CreERT2)*; *Creb5^flox9/flox9* male mouse was mated to a *Creb5^flox9/flox9* female mouse to generate litters containing progeny that were either *Aggrecan1^tm(IRES-CreERT2)*; *Creb5^flox9/flox9* mice or *Creb5^flox9/flox9* mice (which lack the CreERt2 driver allele). Tamoxifen was repeatedly administered to these litters at postnatal days 1–11, to specifically delete Creb5 in postnatal stage articular chondrocytes. Tamoxifen (Sigma, T5648) was dissolved in corn oil at either a concentration of 2 mg/ml or 4 mg/ml. Before postnatal day 7; 50 microliters of 2 mg/ml tamoxifen were used for intragastric injection (delivered on postnatal days 1, 3, 5).

After postnatal day 7; 50 microliters of 4 mg/ml tamoxifen were used for intraperitoneal injection (delivered on postnatal days 7, 9, 11).

### Generation of *Rosa26^iCreb5-HA* mice

*Rosa26^CAGS-LoxStopLox-Creb5-HA/GFP* mice (*Rosa26^iCreb5-HA* mice) were generated following the protocol outlined in ref. 76. The bovine Creb5-HA cDNA was transferred from pENTR-Creb5-HA into the pR26 CAG/GFP Dest destination vector (Addgene # 74281) using Gateway LR Clonase (Thermo Fisher, cat. no. 11791020) to generate the pR26 CAG/LoxStopLox/Creb5-HA-iresGFP targeting vector. The sgRosa26-1 (gRNA) was synthesized (Synthego) and Cas9 protein was purchased from PBA Bio (cat. no. CP01-50). The pR26 CAG/LoxStopLox/Creb5-HA-iresGFP targeting vector and sgRosa26-1 plus Cas9 protein were injected into C57/BL6 one cell mice embryos[74,75] by the Genome Modification Facility at Harvard University. Of 61 pups that were born, 15 animals contained CAG/LoxStopLox/Creb5-HA-iresGFP which had underwent homologous recombination into the *Rosa 26* locus. One of the 15 mice (mouse #11) was mated to C57/BL6 mice to generate the *Rosa26^CAGS-LoxStopLox-CrebSHA-ires-GFP* mice line (termed *Rosa26^iCreb5-HA* mice). Either R26F3 and SAR primers[76] or Creb5F (AGGCGGCGGAAATTTCTG) and Creb5R (CATGGCTGT-TATGGGGCA) primers were used for genotyping (diagram of the *Rosa26^iCreb5-HA* allele is displayed in Supplementary Fig. 4a).

### Generation of *Wnt5a^Δexon2/Δexon2* mice

*Wnt5a^Δexon2/+* mice were generated by crossing a male *Wnt5a^flox-exon2/flox-exon2* mouse[77] (purchased from The Jackson Laboratory, Stock No: 026626) with a female EIIa-cre mouse (purchased from The Jackson Laboratory, Stock No: 003724). The germline deletion of the floxed exon 2 of the *Wnt5a* allele was confirmed by PCR in EIIa-cre negative offspring using the following primers: F: GGTGAGGGACTGGAAGTT and R: TGCTTCA-GACACTGTGGC. The lengths of the PCR products for the wildtype *Wnt5a* allele and the *Wnt5a^Δexon2* allele (after deletion of exon 2) are 832 bp and 330 bp, respectively. *Wnt5a^Δexon2/+* males were crossed with *Wnt5a^Δexon2/+* females to generate litters containing both *Wnt5a^Δexon2/Δexon2* embryos and control embryos (which were either *Wnt5a^Δexon2/+* or *Wnt5a^+/+*). The sequence of the 330 bp PCR product that identifies the *Wnt5a^Δexon2* allele is (primer to primer): GGTGAGGG ACTGGAAGTTGCAGGAAAGAATTCGGCGCGCCATAACGGGGGTATAG GATACATTATACGAAGTTATTTAATTAAGGCTACCATGGAGAAGTTA CTATTCCGAAGTTCCTATTCTCTAGAAAGTATAGGAACTTCAAGCTGA TCCTAAGCTTGGCGCGGCCGCCCGCACCCCCTCACCAGGTAAAAT AAGAGAGAGTAGCCGTATTATTTTTCTGTCGTTTGAGGTTATTAAG CCACAGTGTGTCCCCCAAGGTAAAATAAAAATGAGTTTCCCTTTTA TTTTTCTGTCGTTTGAGGTTATTAAGCCACAGTGTCTGAAGCA.

### Reporting summary

Further information on research design is available in the Nature Portfolio Reporting Summary linked to this article.

## Data availability

All primary RNA-Seq data sets generated and analyzed during the current study are available in the GEO repository with accession number GSE181883. All data and materials used in the analysis will be made available to any researcher for purposes of reproducing or extending the analysis Source data are provided with this paper.

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

## Acknowledgements

We thank Attila Aszodi, Veronique Lefebvre, Maurizio Pacifici and Cliff Tabin for providing in situ probes; Terence Capellini, April Craft, Vicki Rosen, Ramesh Shivdasani, Matt Warman, Yingzi Yang and Cliff Tabin for their constructive comments on the manuscript; Haruhiko Akiyama, Benoit de Crombrugghe and Yingzi Yang for sharing the *Sox9^ires-Cre* mice; Jongkil Kim for help with quantitation of flat cells in the superficial zone of WT and mutant mice; and the Nikon Imaging Center at Harvard Medical School for use of their microscopes and cameras. In addition, we thank the Histology Core Facility at the Center for Skeletal Research at Massachusetts General Hospital (supported by NIH P30 AR066261) for histology recommendations and services; and the Genome Modification Facility at Harvard University for help with CRISPR/Cas9 generation of mouse strains. This work was supported by grants from NIH to A.B.L. (NIAMS: R01AR060735, R01AR074385 and R01AR076562) and by a grant from The Arthritis National Research Foundation to C.-H.Z. (Grant award: 707225). Y.G. was supported by a grant from Ean Technology, Co. Ltd.

## Author contributions

C.-H.Z. and A.B.L. designed experiments and wrote the paper. C.-H.Z. and Y.G. conducted experiments and edited the paper. Z.Z. performed bioinformatics analysis of the RNA-Seq libraries. H.-H.H. isolated superficial and deep zone bovine articular cartilage tissue and edited the paper. A.J.G. provided experimental expertise and edited the paper.

## Competing interests

The authors declare no competing interests.
