## [Peer Review File · Nature Communications]

Creb5 coordinates synovial joint formation with the genesis of articular cartilageReviewer #1 (Remarks to the Author):

The manuscript 'Creb5 is necessary for joint formation and directs the genesis of articular cartilage' by Zhang et al. characterizes the novel role of a transcription factor Creb5 during development of articular cartilage and synovial joints. The authors have developed a transgenic mouse in which the DNA binding domain of Creb5 is deleted and hence its function impaired. The topic is of broad interest as there is limited understanding of how articular cartilage and synovial joints are formed during development, however the manuscript is largely descriptive and is lacking in mechanistic details of Creb5 function in its present form.

Major points:

1. A major caveat in the manuscript is in the characterization of the Creb5 transgenic mouse. There is no data to show that a loss of the exon encoding both the basic region and 3 out of 5 leucines of the Creb5 bZIP DNA-binding domain, actually leads to a loss of Creb5 binding.the manuscript just states that 'it is thus likely to be critical for Creb5 DNA binding'. A chromatin immunoprecipitation (ChIP) experiment in wildtype and mutant chondrocytes, showing a loss of Creb5 binding is critical.
2. Another major caveat is that there is no molecular data showing how Creb5 might affect Gdf5 and Wnt4 expression....is this a direct or an indirect effect?a global ChIP/RNA-seq type of experiment can elucidate the precise effects of Creb5 and identify the target genes.
3. The overlap between Creb5 and Gdf5 in E13.5 embryos appears minimal while at E14.5, the overlap is very strong but the pattern of staining is strikingly different.....this point is not explained or discussed properly... are these cells expressing Prg4 at this point?
4. What might be the temporal targets of Creb5 during the developing stages from E13.5 to P0?

--

Reviewer #2 (Remarks to the Author):

In this work, Zhang and coauthors used different transgenic mice models to investigate the role of the transcription factor Creb5 in the development of articular cartilage and in the formation of cortical bone and growth plates within the developing cartilage elements. The findings are relevant in the field of cartilage development, although they remain rather descriptive. In fact, the manuscript does not provide insights into the mechanisms by which Creb5 promotes fate determination and differentiation of joint cells.

The study would gain scientific strength if it included genetic and/or functional characterization of different cell populations isolated from the developing tissues. For example, gene expression profiling of cells isolated from Creb5+ vs Creb5- perichondria and limb bud mesenchyme tissues would allow to determine whether the morphogenic effects induced by Creb5 are mediated through specific signaling pathways, including wnt and TGFb. In addition, in vitro (co)culture assays would enable to determine the mechanisms by which Creb5-expressing cells modulate the commitment/differentiation of limb bud mesenchymal cells (i.e., through direct cell interactions or by the release of soluble factors).

The authors are also recommended to include quantitative data on the dimension of the limb skeletal elements of the wild type vs transgenic mice to corroborate the histological findings, and to include scale bars in the figures.

--

Reviewer #3 (Remarks to the Author):

This is a very good and important piece of work that describes how the transcription factor Creb5 is required for joint formation and articular cartilage generation. The work follows up a study from the same group (in an article in press also in this journal) in which they identified Creb5 as being expressed in the superficial layer of articular cartilage. In one sense it's a shame the two articles were not combined in one study as this work does essentially extend from the other, however, I can understand the reasons this is not the case.

The authors start by making a new Creb3 null mouse using CRISPR. This removes an essential exon that encodes for part of the DNA binding. A western blot confirming either loss or decrease in protein level should be added. A blot of the overexpression of essentially this version in a cell line is shown but data from the mouse is not provided. These mice, like a full KO, die immediately after birth.

The authors pursue the phenotype of the embryos of these animals with skeletal preps, IHC, and in situ (inc. RNAScope). A brief discussion about why the mice die could be useful. From the mice one finding is that Creb5 is essential for the separation of certain joint elements.

The work then shifts focus to look at another novel mouse where Creb5 is miss-expressed in cells essentially driven by the Prx1 promoter. These mice had suppressed Wnt5a expression in the proximal limb bud perichondrium, which probably subsequently inhibited longitudinal bone growth. The authors then generated a mouse which expressed Creb essentially in cells whenever sox9 was expressed (further detail about this construction is required). Together these mice lead to the conclusion that the extent of expression of Creb5 in the perichondrium correlates with its ability to block both growth plate formation and chondrocyte maturation.

Because there are two 'inducible' Creb5 mouse lines the authors should be careful when using the term 'iCreb5' and clearly state whether they are referring to Prx1-Cre-iCreb5 or Sox9-ires-Cre-iCreb5.

Overall my feeling is that the in situ and descriptions are a little overlong and detailed and, though undoubtedly useful, detract from the main message. Also, but not always presenting controls images adjacent to the experimental one, easy critical assessment is lost. For example, Fig 4D requires similar staining of non-prx1-creb5 driven mice (i.e. WT), or at least the reader needs directing to the comparable figure panel.

My main comment is that although the Prx1-Cre-iCreb5 or Sox9-ires-Cre-iCreb5 mice have provided useful information, perhaps a more beneficial experiment would have been an inducible or tissue specific KO of Creb5.

Minor comments

"ulnohumeral joint" – page 5 refers to 1E not 1D as stated.

Fig 3C – Wnt5a "encroached" into the perichondrium – this is very difficult to see.

More information about the SOX9-ires-Cre-iCreb3 is required. This was presumably generated by reference 33 – but at the moment the nomenclature used and lack of provided details is a little confusing and requires some clarification.

Thanks to all the reviewers for their thoughtful comments on our prior manuscript.

Reviewers' comments:

Reviewer #1 (Remarks to the Author):

The manuscript 'Creb5 is necessary for joint formation and directs the genesis of articular cartilage' by Zhang et al. characterizes the novel role of a transcription factor Creb5 during development of articular cartilage and synovial joints. The authors have developed a transgenic mouse in which the DNA binding domain of Creb5 is deleted and hence its function impaired. The topic is of broad interest as there is limited understanding of how articular cartilage and synovial joints are formed during development, however the manuscript is largely descriptive and is lacking in mechanistic details of Creb5 function in its present form.

Response: Thanks for your interest in our findings. We have added additional data to our revised manuscript (including RNA-Seq following shRNA knockdown of Creb5 to identify Creb5 transcriptional target genes) indicating that Creb5 activates the expression of both the Wnt-antagonist *Wif1* and the atypical cadherins *FAT1/4* in the articular perichondrium. In addition, we present new data indicating that ectopic perichondrial expression of Creb5 disrupts a Wnt5a-Yap/Taz/Tead positive feedback-loop. In terms of mechanism, we have found that Creb5 induces the expression of both *Wif1*, a secreted Wnt5a-antagonist and *Fat4*, an atypical cadherin that can block Yap/Taz/Tead signaling, in both the articular perichondrium as well as in postnatal articular cartilage. Indeed, postnatal deletion of Creb5 in articular cartilage led to increased expression of *Ctgf* in periarticular tissues which lack Creb5 expression; indicating that Creb5 dampens the expression of this Wnt/Yap/Taz transcriptional target in a non-cell autonomous fashion. We also present additional data in the revised manuscript indicating that postnatal deletion of Creb5 in the articular cartilage leads to loss of both flat superficial zone articular chondrocytes coupled with a loss of both *Prg4* and *Wif1* expression; and a non-cell autonomous up-regulation of *Ctgf*. Taken together our findings indicate that Creb5 promotes both joint formation and the subsequent development of articular chondrocytes by disrupting a Wnt5a positive-feedback loop in the perichondrium.

Major points:

1. A major caveat in the manuscript is in the characterization of the Creb5 transgenic mouse. There is no data to show that a loss of the exon encoding both the basic region and 3 out of 5 leucines of the Creb5 bZIP DNA-binding domain, actually leads to a loss of Creb5 binding.....the manuscript just states that 'it is thus likely to be critical for Creb5 DNAbinding' A chromatin immunoprecipitation (ChIP) experiment in wildtype and mutant chondrocytes, showing a loss of Creb5 binding is critical.

Response: In new Supplementary Fig 1. we have added additional molecular characterization of the *Creb5^{Δ9}* allele, which lacks an intact DNA binding domain. In this revised figure, we show a diagram of WT *Creb5* exon9 (blue) and flanking intron sequences depicted with targeting CRISPR guide RNAs. The targeted allele following non-homologous end-joining is depicted and the sequence of the junction in the new chimeric intron that now lies between exon 8 and exon 10 of the *Creb5^{Δ9}* allele is displayed (derived from mouse 8). Consistent with loss of exon 9 in the *Creb5^{Δ9}* allele, RT-PCR of *Creb5* cDNA (derived from brain tissue of either adult *WT* or adult *Creb5^{Δ9/+}* mice) with primers located in *Creb5* exons 8 and 11 indicated that exon 8 is directly spliced (in frame) to exon 10 in the *Creb5^{Δ9}* mRNA. Deletion of exon 9 is predicted to yield an in-

frame mutant Creb5(Δ 9) protein, that specifically lacks 76 amino acids that encode a critical part of the bZIP domain of Creb5 and is necessary for direct interaction of Creb5 with DNA targets. Consistent with this prediction, we have found that while infection of deep zone bovine articular chondrocytes with lentivirus encoding either Creb5(WT)-HA or a mutant form of Creb5 lacking the amino acids encoded by exon 9 (i.e., Creb5(Δ 9)-HA) drove expression of nearly equal levels of HA-tagged protein (Supplementary Fig. 1e), Chromatin-immunoprecipitation (ChIP)-PCR revealed that only Creb5(WT)-HA (and not Creb5(Δ 9)-HA) was able to bind to the enhancers that drive Prg4 expression (Supplementary Fig. 1f). Consistent with the inability of Creb5(Δ 9)-HA to bind to the Prg4 enhancers, we found that while Creb5(WT)-HA was able to induce robust expression of Prg4 in deep zone bovine articular chondrocytes treated with TGF- β 2 and TGF α , Creb5(Δ 9)-HA was unable to induce Prg4 expression (Supplementary Fig. 1g). Taken together, these results indicate that the *Creb5* ^{Δ 9} allele encodes a mutant protein (i.e., Creb5(Δ 9)) which is unable to either bind to regulatory regions in chromatin or induce expression of transcriptional targets via direct protein: DNA interaction.

2. Another major caveat is that there is no molecular data showing how Creb5 might affect Gdf5 and Wnt4 expression....is this a direct or an indirect effect?a global ChIP/RNA-seq type of experiment can elucidate the precise effects of Creb5 and identify the target genes.

Response: We present new data indicating that loss of Creb5 function leads to the loss of expression of both Gdf5, Wif1, Wnt4, Wnt9a, and Wnt11 specifically between the fused joints of *Creb5* ^{Δ 9/ Δ 9} embryos (Figs. 2a, 2e, 2f); and is consistent with prior work indicating the necessity for canonical Wnt signaling in the generation of synovial joints. Interestingly, while Wnt4 expression was lost from joint interzones between the fused cartilage elements, Wnt4 was still expressed in perichondrium located adjacent to the developing tibiofemoral joint in both *WT* and *Creb5* ^{Δ 9/ Δ 9} embryos (Fig. 2d). Taken together, these results indicate that Creb5 is critical to maintain appropriate expression of Gdf5, Wnt-family members and Wif1 in the developing interzones of most (but not all) synovial joints. Thanks for your recommendation to perform a global ChIP/RNA-seq type of experiment to identify Creb5 targets. As recommended by this reviewer, we are currently developing a mouse line (*Creb5*^{3XHA-P2A-tdTomato}/+ mice) which specifically expresses tdTomato in Creb5-HA-expressing cells. We intend to employ *Creb5*^{3XHA-P2A-tdTomato}/+ mice to map the localization of Creb5 on transcriptional targets in the developing synovial joint interzones. However, global analysis of Creb5 target genes will likely take us at least another year to perform; and thus needs to be the topic of a future manuscript. In the present revised manuscript, we do identify Creb5 transcriptional target genes in bovine articular chondrocytes to include Wif1, Gdf10, Fat1, Fat4, EphA3, HBEGF (an EGFR-ligand), and Tgfb1. Consistent with these findings in bovine articular chondrocytes, we similarly find that Creb5 is critical to drive expression of both Wif1 and Fat4 in the developing synovial joint interzones.

3. The overlap between Creb5 and Gdf5 in E13.5 embryos appears minimal while at E14.5, the overlap is very strong but the pattern of staining is strikingly different.....this point is not explained or discussed properly... are these cells expressing Prg4 at this point?

Response: We have added a new RNAscope fluorescent in situ hybridization (FISH) in E13.5 embryos; which indicated that Creb5 is expressed in a broad domain of the developing forelimb autopod, and that its expression in the nascent shoulder and elbow joints significantly overlaps with that of Gdf5 in the interzones of the developing synovial joints (Fig. 1a). By E14.5, Gdf5

expression is similarly restricted to a subset of Creb5 expressing cells in the forelimb autopod (Fig. 1b). Prg4 is not expressed at these early stages of synovial joint formation.

4. What might be the temporal targets of Creb5 during the developing stages from E13.5 to Po?

Response: Thanks for this excellent question! One of our plans for the *Creb5*^{3XHA-P2A-tdTomato}/+ mice are to specifically identify the transcriptional targets that Creb5 activates during differing stages of synovial joint development. This analysis will likely take us at least a year to perform, and thus needs to be the topic of a future manuscript.

--

Reviewer #2 (Remarks to the Author):

In this work, Zhang and coauthors used different transgenic mice models to investigate the role of the transcription factor Creb5 in the development of articular cartilage and in the formation of cortical bone and growth plates within the developing cartilage elements. The findings are relevant in the field of cartilage development, although they remain rather descriptive. In fact, the manuscript does not provide insights into the mechanisms by which Creb5 promotes fate determination and differentiation of joint cells.

Response: We have added additional data to our revised manuscript, including RNA-Seq following shRNA knockdown of Creb5 in bovine articular chondrocytes, to identify Creb5 transcriptional target genes. This analysis revealed that Creb5 activates the expression of both the Wnt-antagonist *Wif1* and the atypical cadherins *FAT1/4* in both articular chondrocytes and in the articular perichondrium. In addition, we present new data indicating that ectopic perichondrial expression of Creb5 disrupts a *Wnt5a-Yap/Taz/Tead* positive feedback-loop. In terms of mechanism, we have found that Creb5 induces the expression of both *Wif1*, a secreted Wnt5a-antagonist and *Fat4*, an atypical cadherin that can block *Yap/Taz/Tead* signaling, in both the articular perichondrium as well as in postnatal articular cartilage. Indeed, postnatal deletion of Creb5 in articular cartilage led to increased expression of *Ctgf* in periarticular tissues which lack Creb5 expression; indicating that Creb5 dampens the expression of this Wnt/*Yap/Taz* transcriptional target in a non-cell autonomous fashion. We also present additional data in the revised manuscript indicating that postnatal deletion of Creb5 in the articular cartilage leads to loss of both flat superficial zone articular chondrocytes coupled with a loss of both *Prg4* and *Wif1* expression; and a non-cell autonomous up-regulation of *Ctgf*. Taken together our findings indicate that Creb5 promotes both joint formation and the subsequent development of articular chondrocytes by disrupting a *Wnt5a* positive-feedback loop in the perichondrium.

The study would gain scientific strength if it included genetic and/or functional characterization of different cell populations isolated from the developing tissues. For example, gene expression profiling of cells isolated from Creb5+ vs Creb5- perichondria and limb bud mesenchyme tissues would allow to determine whether the morphogenic effects induced by Creb5 are mediated through specific signaling pathways, including wnt and TGFb. In addition, in vitro (co)culture assays would enable to determine the mechanisms by which Creb5-expressing cells modulate the commitment/differentiation of limb bud mesenchymal cells (i.e., through direct cell interactions or by the release of soluble factors).

Response: In the revised manuscript we identify Creb5 transcriptional target genes in bovine articular chondrocytes to include *Wif1*, *Gdf10*, *Fat1*, *Fat4*, *EphA3*, *HBEGF* (an EGFR-ligand), and *Tgfb1*; thus suggesting that Creb5 coordinates several distinct signaling pathways to maintain

the health of the articular cartilage. In addition, we also present data in the revised manuscript indicating that postnatal deletion of *Creb5* in the articular cartilage leads to loss of both flat superficial zone articular chondrocytes coupled with a loss of both *Prg4* and *Wif1* expression; and a non-cell autonomous up-regulation of *Ctgf*. Thanks for your excellent recommendation to perform an RNA-seq type of experiment to identify *Creb5* targets. As recommended by this reviewer, we are currently developing a mouse line (*Creb5^{3XHA-P2A-tdTomato}/+* mice) which specifically expresses tdTomato in *Creb5*-HA-expressing cells. We intend to employ *Creb5^{3XHA-P2A-tdTomato}/+* mice to map the localization of *Creb5* on transcriptional targets in the developing synovial joint interzones. However, global analysis of *Creb5* target genes will likely take us at least another year to perform; and thus needs be the topic of a future manuscript.

The authors are also recommended to include quantitative data on the dimension of the limb skeletal elements of the wild type vs transgenic mice to corroborate the histological findings, and to include scale bars in the figures.

Response: Thanks for the recommendation. We have added scale bars to all of our microscopy images.

--

Reviewer #3 (Remarks to the Author):

This is a very good and important piece of work that describes how the transcription factor *Creb5* is required for joint formation and articular cartilage generation. The work follows up a study from the same group (in an article in press also in this journal) in which they identified *Creb5* as being expressed in the superficial layer of articular cartilage. In one sense it's a shame the two articles were not combined in one study as this work does essentially extend from the other, however, I can understand the reasons this is not the case. The authors start my making a new *Creb3* null mouse using CRISPR. This removes an essential exon that encodes for part of the DNA binding. A western blot confirming either loss or decreases mw should be added. A blot of the overexpression of essentially this version in a cell line is shown but data from the mouse is not provided. These mice, like a full KO, die immediately after birth. The authors pursue the phenotype of the embryos of these animals with skeletal preps, IHC, and in situ (inc. RNAScope). A brief discussion about why the mice die could be useful. From the mice one finding is that *Creb5* is essential for the separation of certain joint elements.

Response: Thank you for your kind words about our prior submitted manuscript. We tried to visualize *Creb5* WT versus *Creb5*($\Delta 9$) protein by Western blot of mouse tissues; but thus far, we have been unsuccessful (with the commercially available *Creb5* antibodies) to be able to detect endogenous *Creb5* in mouse tissues by Western blot. On the other hand, we present extensive analysis of both the genomic sequences of the *Creb5 ^{$\Delta 9$}* allele and mRNA encoded by this gene, indicating that the *Creb5 ^{$\Delta 9$}* allele encodes a transcript in which exon 8 is directly spliced (in frame) to exon 10 in the *Creb5 ^{$\Delta 9$}* mRNA. Deletion of exon 9 is predicted to yield an in-frame mutant *Creb5*($\Delta 9$)- protein, that specifically lacks 76 amino acids that encode a critical part of the bZIP domain of *Creb5* and is necessary for direct interaction of *Creb5* with DNA targets. Consistent with this prediction, we have found that while infection of deep zone bovine articular chondrocytes with lentivirus encoding either *Creb5*(WT)-HA or a mutant form of *Creb5* lacking the amino acids encoded by exon 9 (i.e., *Creb5*($\Delta 9$)-HA) drove expression of nearly equal levels of HA-tagged protein (Supplementary Fig. 1e), Chromatin-immunoprecipitation (ChIP)-PCR revealed that only *Creb5*(WT)-HA (and not *Creb5*($\Delta 9$)-HA) was able to bind to the enhancers that drive *Prg4* expression (Supplementary Fig. 1f). Consistent with the inability of *Creb5*($\Delta 9$)-HA to

bind to the Prg4 enhancers, we found that while Creb5(WT)-HA was able to induce robust expression of Prg4 in deep zone bovine articular chondrocytes treated with TGF- β 2 and TGF α , Creb5(Δ 9)-HA was unable to induce Prg4 expression (Supplementary Fig. 1g). Taken together, these results indicate that the *Creb5* ^{Δ 9} allele encodes a mutant protein (i.e., Creb5(Δ 9)) which is both unable to either bind to regulatory regions in chromatin or induce expression of transcriptional targets via direct protein: DNA interaction. While we were unable to detect endogenous Creb5 WT versus Creb5(Δ 9) protein by Western blot of mouse tissues, we were able to employ a commercially available antibody to detect expression of both WT and mutant proteins by immunofluorescence in the articular perichondrium of either Po *Creb5* ^{Δ 9/ Δ 9} mice or their control littermates (Fig. 3). At present, do not know why the *Creb5* ^{Δ 9/ Δ 9} mice die after birth, as we have focused our efforts at understanding the role of Creb5 in the developing synovial joints. We will however explore the role of Creb5 in other tissues in future analyses.

The work then shifts focus to look at another novel mouse where Creb5 is miss-expressed in cells essentially driven by the Prx1 promoter. These mice had suppressed Wnt5a expression in the proximal limb bud perichondrium, which probably subsequently inhibited longitudinal bone growth. The authors then generated a mouse which expressed Creb essentially in cells whenever sox9 was expressed (further detail about this construction is required). Together these mice lead to the conclusion that the extent of expression of Creb5 in the perichondrium correlates with its ability to block both growth plate formation and chondrocyte maturation. Because there are two 'inducible' Creb5 mouse lines the authors should be careful when using the term 'iCreb5' and clearly start whether they are referring to Prx1-Cre-iCreb5 or Sox9-ires-Cre-iCreb5.

Response: Thank you for this recommendation. We now refer to iCreb5 as being either iCreb5-HA^(Prx1-Cre) or iCreb5-HA^(Sox9-Cre).

Overall my feeling is that the in suit and descriptions are a little overlong and detailed and, though undoubtably useful, detract from the main message. Also, but not always presenting controls images adjacent to the experimental one, easy critic or assessment is lost. For example, Fig 4D requires similar staining of non-prx1-creb5 driven mice (i.e. WT), or at least the reader needs directing to the comparable figure panel.

Response: We have tried to present a thorough description of the synovial joint phenotypes in the various mouse strains we discuss. We have taken pains to discuss these phenotypes in detail, as we thought this information may be useful for future work (by others) with these animals. We have tried to focus our discussion on the most important points we present. In Supplementary Fig 4. we document that ectopic expression of iCreb5-HA^(Prx1-Cre) can both boost the expression of interzone signaling molecules and repress expression of Wnt5a in the perichondrium. In this revised figure, gene expression assayed by FISH in serial sections of the forelimb of an E14.5 *Prx1-Cre-iCreb5* embryo is compared to that of a control littermate.

My main comment is that although the Prx1-Cre-iCreb5 or Sox9-ires-Cre-iCreb5 mice have provided useful information, perhaps a more beneficial experiment would have been an inducible or tissue specific KO of Creb5.

Response: Thank you for this excellent recommendation! To study the role of Creb5 in postnatal synovial joints we generated mice containing Creb5 alleles in which exon 9 (which encodes the bZIP DNA binding domain) is flanked by loxP sites (*Creb5*^{lox9/lox9} mice; Supplementary Figs. 5a & 5b). Like *Creb5* ^{Δ 9/ Δ 9} animals, *Prx1-Cre; Creb5*^{lox9/lox9} mice also displayed an absence of a knee joint (Supplementary Fig. 5c), indicating that the floxed alleles in *Creb5*^{lox9/lox9} mice can be deleted by Cre-mediated recombination. We mated *Creb5*^{lox9/lox9} mice with *Aggrecan*^{tm(IRES-}

CreERT2; Creb5^{flox9/flox9} mice to generate litters containing both *Aggrecan1^{tm(IRES-CreERT2); Creb5^{flox9/flox9}}* mice or *Creb5^{flox9/flox9}* control mice (which lack the CreERT2 driver allele). Tamoxifen was repeatedly administered to these litters at postnatal days 3-13, to specifically delete *Creb5* in postnatal stage articular chondrocytes. The animals were sacrificed at two weeks of age, and their hindlimbs were cryo-sectioned for RNAscope analysis. We observed that expression of both *Prg4* and *Wif1* was markedly decreased in P14 *Aggrecan1^{tm(IRES-CreERT2); Creb5^{flox9/flox9}}* mice in comparison to their control *Creb5^{flox9/flox9}* littermates (Fig. 10a). In striking contrast to the loss of both *Prg4* and *Wif1* expression following deletion of *Creb5* in the forming joint, the expression of the Wnt/Yap/Taz/Tead target, *Ctgf*, was increased in both periarticular chondrocytes and in the metaphyseal perichondrium (Fig. 10a). The knees of either P21 or P42 *Aggrecan1^{tm(IRES-CreERT2); Creb5^{flox9/flox9}}* mice or *Creb5^{flox9/flox9}* mice were sectioned and stained with Safranin O/ Fast Green, to assay the histology of the articular cartilage. We observed that superficial zone flat cells on the surface of the articular cartilage were absent in the knees of P21 *Creb5*-deficient mice, and that by P42, cells in the interior of the meniscus displayed premature ossification in these animals (Fig. 10b).

Minor comments

“ulnohumeral joint” – page 5 refers to 1E not 1D as stated.

Response: This image is currently shown in Figure 2a. I believe that we see residual expression of both *Gdf5* and *Wnt4* in the developing ulnohumeral joint of *Creb5^{Δ9/Δ9}* mice, which is apparent in the lower panels of this figure.

Fig 3C – *Wnt5a* “encroached” into the perichondrium – this is very difficult to see.

Response: In New Figure Fig 4, we repeated this analysis with multiplexed RNA-Scope FISH to better document that *Creb5* is expressed in a reciprocal pattern to *Wnt5a* in the perichondrium. In this new figure, gene expression of *Creb5* and *Wnt5a* was assayed by multiplexed RNAscope FISH in the forelimb of a WT E14.5 mouse embryo (a) or the knee of an E14.5 *Creb5^{Δ9/Δ9}* embryo or control littermate (b). In WT/control limbs, the articular perichondrium (yellow arrows) and metaphyseal/diaphyseal perichondrium (white arrows) are indicated. In *Creb5^{Δ9/Δ9}* mutant limb, the perichondrial cells expressing both *Wnt5a* and mutant *Creb5^{Δ9/Δ9}* (white arrowheads) are indicated.

More information about the *SOX9-ires-Cre-iCreb3* is required. This was presumably generated by from reference 33 – but at the moment the nomenclature used and lack of provided details is a little confusing and requires some clarification.

Response: Thanks for this recommendation. We have added the appropriate citation for these animals, “we mated *Sox9^{ires-Cre}* mice ³⁷ with *Rosa26^{iCreb5-HA}* mice to generate *Sox9-ires-Cre-iCreb5* embryos, which expressed *iCreb5-HA^(Sox9-Cre)* in most chondrocytes and in a few perichondrial cells (Fig. 7d)”

Reviewer #1 (Remarks to the Author):

The authors have suitably addressed all of the prior critiques.

Reviewer #3 (Remarks to the Author):

In this revision the authors have improved the manuscript. However, in general the article is overlong and somewhat observational.

The deletion of exon 9 and loss of function of Creb5 remains the most interesting aspect of the work. The new (supplemental data) is useful confirming loss of function, but again I'm unclear why there is no western blot – the authors have done creb westerns in ref. 18.

The artificial over/miss-expression of Creb sections are particularly over long and need severe editing. The findings are nicely summed with the statement "Our findings suggest that iCreb5-HA(Prx1-Cre) expression in limb bud mesenchymal cells both promotes the generation of an articular perichondrium and blocks formation of both cortical bone and growth plates within the forming cartilage elements". The authors should find a way of summing the data on this mouse into one figure.

Figure references are not always in order, which is annoying for the reader.

Where possible arrows should be used to point the reader to the text description of the data, - for example, fig2b – distinct elements need indicating in controls, feature of interest are very hard to distinguish – and will be for a non-expert in limb development.

The statement "We observed that superficial zone flat cells on the surface of the articular cartilage were absent in the knees of P21 Creb5-deficient mice" needs some form of quantitation. Was cartilage thickness reduced?

The Crebflox mouse needs more analysis/ controls. Mouse 3 for example (supp fig 5b) appears to contain 1 wt and 1 loxp allele?

Some functional work establishing the role of creb5 in the wnt5/yap/taz loop, otherwise the text is merely speculative.

Reviewer #4 (Remarks to the Author):

The manuscript entitled 'Creb4 is necessary for joint formation and directs the genesis of articular cartilage' by Zhang et al. describes the functions of the transcription factor Creb5 in joint formation during embryonic development using a global knock-out allele and a conditional transgene allowing for tissue-specific Creb5 expression. In addition, the authors provide data on the postnatal function of Creb5 using the TAM-inducible Agg-CreERT2 line. Overall this is an interesting manuscript suggesting that Creb5 plays a role in the formation of the joint interzone and is postnatally required for the maintenance of the superficial layer of the articular chondrocytes. In this manuscript, the authors have also incorporated most of the previous concerns raised by the reviewers. Yet, there are still a few major and minor concerns regarding the interpretation of the results and the mechanism proposed.

Furthermore, the figures need some more work (see comments regarding the figures below).

Comments:

- A major concern is that the embryonic skeletal phenotypes described here, in the Creb5 Δ 9/ Δ 9 mice, could be in part be due to defects in muscles. This is a total knock-out and Creb5 has also been reported in the literature to be expressed in the musculature and from the images shown here for Creb5 transcript and protein, it appears that there is some staining in the soft tissue as well. Furthermore, it is known from the literature that spontaneous muscle contractions are required for proper joint formation during embryonic development.

- As for the transgenic study expressing Creb5 under the control of Prx1Cre, it is surprising that the transgene expression appears rather limited to certain areas and not as expected throughout the entire limb mesenchyme. Immuno-stained sections for the HA-tag of the transgene should be provided.

- Based on the whole-mount skeletal preparations of the transgenic animals provided the skeletal elements seemed to be pretty messed up. Hence, in sectioned material, misinterpretations due to the level of sectioning are possible. Unfortunately, some of the sections from the transgenic limbs may not be oriented properly, and hence, some of the skeletal elements may not be cut in the most optimal way. Most relevant, the plane of the section can have an influence on the marker staining. For e.g. the proximal element (humerus) is not visible in the control and only (the distal) part of it appears to be visible in the Prx1-Cre-iCreb5 sample in Figure 5a. Nevertheless, the authors use this example to claim that Wnt5a is no longer expressed in the perichondrium of the proximal element. Furthermore, the sample shown in 5b, looks quite different - why is there no Wnt5a signal (to the same extent) visible in the perichondrium surrounding the distal elements as in 5a (white arrowhead). Additional perichondrial markers should be used (Bandyopadhyay et al. 2008) to show that the effect is specific to Wnt5a (as claimed by the authors).

- On some of the images of Prx1-Cre-iCreb5 samples, it appears as if there are more cartilage elements than anticipated (see for example Col2 staining in Suppl. Fig. 4). This is also apparent in the whole-mount skeletal preparation shown in Figure 6a. Therefore, it would be helpful if the authors could indicate which skeletal elements are seen on the individual images and to which joint the arrowhead is pointing.

As chondrogenesis appears to be severely altered and delayed in the Prx1-Cre-iCreb5 limbs, it is very difficult to make statements regarding the intensity of signals such as Gdf5 and Wnt4 compared to a control that is more developmentally advanced. Hence, the authors would need to provide complementary data to show that the levels of Gdf5 and Wnt4 are elevated in the Prx1-Cre-iCreb5 limbs.

- How do the authors explain the polydactyly phenotype (which to a certain extent is also visible in the forelimb and not just in the hindlimb)? As pointed out in other places, it appears that additional skeletal elements are also visible in the more proximal region (Figure 6a, Col2 staining in Suppl. Fig. 4). This should be documented better and may argue in support of the idea that Creb5 can induce an interzone-like structure leading to the segmentation of skeletal elements. For this, it would probably be necessary to look in earlier limbs (such as E11.5-E13.5) and also document the expression of the transgene using the HA-tag better.

- Limb bud micromass cultures could be performed. Here, the expectation would be that Prx1-Cre-iCreb5 limbs should give rise to smaller nodules. In this culture, the localization of the HA-tagged transgene should also be analyzed. Based on the hypothesis of the authors it should accumulate in cells surrounding the cartilage nodules.

- The data on the Sox9-IRES-Cre are convincing. Interestingly, bone protrusions such as the deltoid tuberosity and the olecranon appear to be missing or are reduced in these mutants. Whether the patella is also affected cannot be judged from the image provided in Figure 7c.

- The differences in Fat4 and Wif1 expression patterns in the joint interzone of the Creb5 Δ 9/ Δ 9 mutants do not really support the conclusion by the authors that Creb5 promotes the expression of these two genes. As the joint does not form normally genes expressed in that region can be altered simply as a consequence. The same is true for the in vitro experiments using shRNA for Creb5. This may also lead to an alteration of the cell type. Promotor studies would be needed to show a direct interaction. How do the authors explain the differences in the recovery of the deregulated genes between conditions A and AB.

Minor comments:

- Please mention also briefly in the result part, when introducing the mutant allele, that the deletion of exon 9 leads to an in-frame deletion in the CREB5 protein.

- On page 7, the authors describe that the CREB5 Δ 9 protein can still be detected in the radiohumeral joint and so can phospho-Smad2. Is this observed in all joints or specific for this particular joint? In the knee joint of Creb5 Δ 9/ Δ 9 specimens (shown in Figure 2d-f), this does not seem to be the case for CREB5 Δ 9. This needs to be discussed.

- Page 8: The concluding sentence 'Thus, not only is Creb5 activity necessary to promote the expression of joint interzone markers, but it is also critical to exclude the expression of Wnt5a from the perichondrium adjacent to the forming joint' is a bit of an overstatement, as it has not been shown that Creb5 activity promotes the expression of any joint marker except for Prg4 which is a rather late marker and not really an interzone marker. Given that the architecture of the joint is lost the abnormal expression of Wnt5a could just be a consequence thereof. Yet, the authors make it sound as if Creb5 activity is directly involved in the regulation of interzone marker genes and of Wnt5a.

- Please indicate how many independent samples have been analyzed to confirm the staining results.
- Why were deep zone chondrocytes used for the PRG4 Chip experiments. Prg4 is normally a marker for superficial chondrocytes and the latter was used for the RNAseq experiments.
- Page 9: the statement 'Consistent with the requirement for Wnt5a to induce expression of the osteogenic transcription factor Runx2 in the perichondrium (35)...' is incorrect. The paper by Yang et al. 2003 does not show that Wnt5a is required for Runx2 expression. The paper only shows that chondrocyte maturation is delayed in Wnt5a mutants and as a consequence Runx2 expression and osteoblast differentiation are delayed. Accordingly, chondrocyte differentiation may also be delayed in the Prx1-Cre-iCreb5 limbs. This should be analyzed using markers such as Col10a1 and Ihh. Especially the latter has been shown to couple chondrogenesis to osteoblast differentiation in the surrounding perichondrium/periosteum.
- Figures should be organized in the way they are mentioned in the main text. This is not the case for Figures 6d and 5d, which are discussed after Figure 7. Figure 5d even after the mentioning of Figure 6d.
The images shown in Figure 6d are presumably from a forelimb sample (as stated in the figure legend). Yet, this sample shows clear polydactyly and is, therefore, more likely derived from a hindlimb. Furthermore, the elements shown here appear more likely to be carpal or tarsal elements, which would also be surrounded by Prg4 and Wif1 positive cells. What the Col2a1/Matn1 staining would look like on carpal or tarsal elements would need to be analyzed.
- Regarding the Wnt5a Δ ex2 allele, the authors should provide RT-PCR data showing that this is not a transcript null.
- The supplementary tables 1 and 2 are currently provided as excel files. These excel files need to be reorganized into proper columns and the different treatment conditions need to be indicated. Currently, it is not possible to see which excel corresponds to which treatment.
- Images of E14.5 knee joints should be provided for the conditional Prx1-Cre mediated knock-out of Creb5, as these allow better comparison with the straight knock-out.
- RNAscope analysis using a probe directed against exon9 of Creb5 should be performed to show deletion of Creb5 in the postnatal sample. Images showing postnatal expression of Creb5 in the joints should also be provided.
- The layer of flat chondrocytes appears to be still present in the articular cartilage of the upper skeletal element (distal end of the femur?) in the AggCreERT2;Creb5fl/fl samples. Please indicate in the images in Figures 10a and b what is the femur and what tibia. The articular cartilage in the P42 samples of controls and AggCreERT2;Creb5fl/fl samples does not appear to be very different.
- According to the publication by Gamer et al. 2016 the anterior (4 weeks) and posterior menisci (8 weeks) mineralize at different timepoints in mice. Hence, it would be good if the authors make sure that they compared the same menisci (currently there is no information provided in the figure legend on which part of the meniscus is shown). Images of sections in which both menisci are visible may be more useful here (or micro CT images of the whole knee).
- Global misexpression of Wnt5a in chicken limbs (Hartmann and Tabin, 2000) or throughout the Col2a1-expressing domain (including chondrocytes at the joints; Yang et al. 2003) did not lead to joint alterations as would be expected based on the model of the mechanism proposed here. Can the authors explain this?
- Regarding the proposed hypothesis that Creb5 in the articular perichondrium may influence the timing of ossification, the authors should provide evidence that the onset of SOC formation is altered in conditional Prx1Cre-Creb5fl/fl specimens. Currently, this is highly speculative and based on an embryonic phenotype that can be interpreted in alternative ways.
- Some of the images shown in Figure 3c are identical to images shown in Figure 2e from the previous publication by Zhang et al. 2021 and hence, may not correspond to littermate controls.
- In their previous paper, the authors already provided experimental data using the iCreb5HA construct that Creb5 can in combination with TGFbeta2 and TGFalpha upregulate Prg4 expression. Yet, relative expression levels in the previous experiments were increased in the presence of TGFbeta2 alone only about 18-fold, and here around 70-fold, while cotreatment with TGFbeta2 and TGFalpha previously led to an increase in the levels by about 44-fold (here almost 80-fold). In both experiments, only two independent samples have been analyzed. The huge discrepancy between these independent experiments is slightly alarming.

Remarks regarding the figures:

- It would be helpful if the authors could reorganize the figures in a way that the limb structures in all images are always orientated in the same way throughout all figures (like proximal to the left and distal to the right or vice versa).
- Also labeling should be more consistent: same font size, size bars either in all images or always in the right-side panels (if all images of that panel are taken at the same magnification), embryonic stage, and anatomical region should be consistently indicated within a figure.
- Additional labels would be helpful within the figures pointing out the specific structures mentioned in the text when describing the phenotype as for non-experts some of the described changes may be difficult to spot.
- Please indicate in all figure legends which skeletal elements or limbs are shown. This is for example not clear from Figure 5a. Furthermore, regarding Fig. 5a, why is the staining for the control in the left panel (Gdf5) different from the Prx1-Cre-iCreb5 sample (iCreb5). What is the genotype of the control littermates in this case? The overall quality of Fig. 5a is poor as the skeletal elements are very difficult to see. The high perichondrial expression of the transgene is also not visible in that image. Maybe the authors can provide immuno-stained images using the HA tag to visualize specifically the expression of the transgene in these samples.
- Figure 6c: Here it would be better to pick regions from the skeletal element to the left (which contains a mineralized region) instead of showing regions from the skeletal element which may all be tarsal-like bones and hence would look also undifferentiated in the control.
- Suppl. Fig. 2c: Prg4 staining in the forelimb is shown in a phalangeal element not in carpal elements as suggested by the title. Carpal elements should be labeled so that the reader can appreciate the joint alterations of certain carpal elements. Also, the tarsal elements should be labeled, as currently, it is difficult to actually see which elements are shown in the images.
- High magnification images of the distinct regions indicated by the arrows and arrowheads would be helpful.

Additional comments:

Some of the histological images lack scale bars (e.g. Figure 3)

In some places, it looks like 'P0' is spelled 'Po' (e.g. page 6).

Spelling mistake on page 13 and in the legend to Suppl. Table 2: TFGalpha instead of TGFalpha

Thanks to our reviewers for their thoughtful comments. Our responses are highlighted in yellow.

REVIEWER COMMENTS

Reviewer #1 (Remarks to the Author):

The authors have suitably addressed all of the prior critiques.

Thanks!

Reviewer #3 (Remarks to the Author):

In this revision the authors have improved the manuscript. However, in general the article is overlong and somewhat observational. The deletion of exon 9 and loss of function of Creb5 remains the most interesting aspect of the work. The new (supplemental data) is useful confirming loss of function, but again I'm unclear why there is no western blot – the authors have done creb westerns in ref. 18.

We tried to visualize Creb5 WT versus Creb5(Δ 9) protein by Western blot of mouse tissues; but thus far, we have been unsuccessful (with the commercially available Creb5 antibodies) to be able to detect endogenous Creb5 in mouse tissues by Western blot. The successful Creb5 westerns that we previously published (in ref. 18) were performed with isolated superficial zone bovine articular chondrocytes, which express relatively high levels of these protein. Using this same antibody, we have been unable to detect this protein in either whole adult mouse brains or in whole limb bud tissues from newborn mice. On the other hand, we present extensive analysis of both the genomic sequences of the *Creb5* ^{Δ 9} allele and mRNA encoded by this gene, indicating that the *Creb5* ^{Δ 9} allele encodes a transcript in which exon 8 is directly spliced (in frame) to exon 10 in the *Creb5* ^{Δ 9} mRNA. Deletion of exon 9 is predicted to yield an in-frame mutant Creb5(Δ 9)- protein, that specifically lacks 76 amino acids that encode a critical part of the bZIP domain of Creb5 and is necessary for direct interaction of Creb5 with DNA targets. Consistent with this prediction, we have found that while infection of deep zone bovine articular chondrocytes with lentivirus encoding either Creb5(WT)-HA or a mutant form of Creb5 lacking the amino acids encoded by exon 9 (i.e., Creb5(Δ 9)-HA) drove expression of nearly equal levels of HA-tagged protein (Supplementary Fig. 1e), Chromatin-immunoprecipitation (ChIP)-PCR revealed that only Creb5(WT)-HA (and not Creb5(Δ 9)-HA) was able to bind to the enhancers that drive *Prg4* expression (Supplementary Fig. 1f). Consistent with the inability of Creb5(Δ 9)-HA to bind to the *Prg4* enhancers, we found that while Creb5(WT)-HA was able to induce robust expression of *Prg4* in deep zone bovine articular chondrocytes treated with TGF- β 2 and TGF α , Creb5(Δ 9)-HA was unable to induce *Prg4* expression (Supplementary Fig. 1g). Taken together, these results indicate that the *Creb5* ^{Δ 9} allele encodes a mutant protein (i.e., Creb5(Δ 9)) which is both unable to either bind to regulatory regions in chromatin or induce expression of transcriptional targets via direct protein: DNA interaction. While we were unable to detect endogenous Creb5 WT versus Creb5(Δ 9) protein by Western blot of mouse tissues, we were able to employ a commercially available antibody to detect expression of both WT and mutant proteins by immunofluorescence in the articular perichondrium of either Po *Creb5* ^{Δ 9/ Δ 9} mice or their control littermates (Fig. 3c).

The artificial over/miss-expression of Creb sections are particularly over long and need severe editing. The findings are nicely summed with the statement “Our findings suggest that iCreb5-HA(Prx1-Cre) expression in limb bud mesenchymal cells both promotes the generation of an

articular perichondrium and blocks formation of both cortical bone and growth plates within the forming cartilage elements”. The authors should find a way of summing the data on this mouse into one figure.

We have tried to concisely describe our results with the *Prx1-Cre-iCreb5* animals, however we have not been able to reduce the number of figures, as we think that these images all make important contributions to our manuscript. Indeed, in response to Review #4's questions we have now included additional data regarding these animals, in which we address whether *iCreb5-HA^(Prx1-Cre)* affects the expression of other perichondrial markers (such as *Crabp1*) and present further evidence that expression of several joint interzone markers (including *Wnt4*, *Wnt11* and *Gdf5*) were each expanded in the perichondrium surrounding the stylopod and zeugopod cartilage elements of E14.5 *Prx1-Cre-iCreb5* embryos. Reviewer #4 also asked that we address how *iCreb5-HA^(Prx1-Cre)* induces polydactyly, as the hindlimb autopods of P0 *Prx1-Cre-iCreb5* mice displayed polydactyly and an excessive number of tarsal elements (Fig. 6a). To understand what might cause these phenotypes, we examined both the size of the limb buds; and the expression of *Msx1* in the hindlimb buds of E11.5 *Prx1-Cre-iCreb5* embryos and control littermates. We noted that both forelimb and hindlimb buds were larger in E11.5 *Prx1-Cre-iCreb5* embryos versus their control littermates (Supplementary Fig. 5a). Prior work has established that BMP signaling promotes the expression of *Msx1* in the limb bud mesenchyme, can drive apoptosis of limb bud mesenchymal cells, and also restricts the expression of *Fgf4/8* in the apical ectodermal ridge (AER). Indeed, manipulations that decrease the level of BMP signaling in the limb bud both promote an expansion of *Fgf4/8* (and the AER) into a larger region of the limb bud ectoderm and induce polydactyly. Interestingly, we observed that misexpression of *iCreb5-HA^(Prx1-Cre)* in E11.5 embryos both expanded the expression of *Gdf5* to a broader domain of the limb bud; and simultaneously decreased the expression of *Msx1* in the limb bud mesenchyme (Supplementary Fig. 5c). As *Gdf5* has been found to block signaling by *BMPR1a*, we speculate that both the increased size of limb buds in E11.5 *Prx1-Cre-iCreb5* embryos, repression of *Msx1*-expression and hindlimb polydactyly may all reflect an attenuation of BMP-signaling in the limb bud mesenchyme due to ectopic expression of *Gdf5*.

Figure references are not always in order, which is annoying for the reader.

Where possible arrows should be used to point the reader to the text description of the data, - for example, fig2b – distinct elements need indicating in controls, feature of interest are very hard to distinguish – and will be for a non-expert in limb development.

Thank you for this recommendation. We have tried our best to arrange the figures as they are discussed in the text. We have also added arrows; indicate the identity of the cartilage/bone elements in some of our figures; and have made a point to indicate both the anatomical location and age of the embryo/mouse that is displayed in each figure.

The statement “We observed that superficial zone flat cells on the surface of the articular cartilage were absent in the knees of P21 *Creb5*-deficient mice” needs some form of quantitation. Was cartilage thickness reduced?

In parallel with the loss of *Prg4* expression in the superficial cells of the articular cartilage in *Aggrecan1^{tm(IRES-CreERT2);Creb5^{flox9/flox9}}* mice animals (Fig. 10a), we similarly observed that superficial zone flat cells on the surface of either the articular cartilage or the meniscus were largely absent from the knees of these animals (Figs. 9c & 10a). To quantify both the location and morphology of *Col2a1*-expressing cells in the articular cartilage we performed RNAScope analysis (Fig. 10b). We noted that postnatal loss of *Creb5* function in the articular cartilage led to

approximately a 75% decline in the number of Col2a1-expressing flat cells in the superficial-most two cell layers of the articular cartilage, on both the femoral and tibial surface in the knees of P30 *Aggrecan1^{tm(IRES-CreERT2); Creb5^{lox9/lox9}}* mice (Fig. 10c). In addition to altering both *Prg4* and *Wif1* expression in the articular cartilage itself, we also noted that postnatal loss of *Creb5* function resulted in a decreased number of *Col10a1* expressing cells on the side of the secondary ossification center that is nearest the articular surface in P30 *Aggrecan1^{tm(IRES-CreERT2); Creb5^{lox9/lox9}}* mice (Fig. 9b). This finding suggests that *Creb5* (in the articular cartilage) induces the expression of signaling molecules that promote the maintenance of *Col10a1* expressing cells in the adjacent secondary ossification center and blocks either their premature loss and/or developmental maturation into osteocytes. Consistent with the notion that *Creb5* indirectly blocks premature endochondral ossification, we also observed premature hypertrophy (in P30 mice) and subsequent endochondral ossification (in P42 mice) of the anterior meniscus in *Aggrecan1^{tm(IRES-CreERT2); Creb5^{lox9/lox9}}* mice that been treated with tamoxifen (Fig. 9c).

The *Creb5* mouse needs more analysis/ controls. Mouse 3 for example (supp fig 5b) appears to contain 1 wt and 1 loxp allele?

The PCR analysis displayed in New Supplementary Figure 9b was performed with genomic DNA derived from the tails of founder *Creb5^{lox}* mice, which were generated following injection of Cas9 protein and guide RNAs into 1 cell mice embryos. As the resultant CRISPR/Cas9 directed gene mutations sometimes do not occur prior to the first cell division, the resultant founder mice frequently are composed of both cells which have been genetically altered along with unaltered WT cells. Indeed, PCR performed with genomic DNA derived from the tails of the founder *Creb5^{lox9}* mice indicated that their tail DNA contained both genetically altered *Creb5* loci and unaltered WT *Creb5* loci. Tail DNA isolated from founder mouse 3 contained both a genetically altered *Creb5* allele with LoxP sites on both sides of exon9 (amplified by the designated primers to yield both 377 bp and 358 bp PCR fragments) and in addition contained a *Creb5* allele lacking the rightward LoxP site (amplified by F2 and R2 primers to yield a 318 bp PCR fragment). However, after mating this same *Creb5^{lox9}* founder mouse 3 with a C57Bl6 mouse, we were able to establish a mouse line in which the *Creb5^{lox9}* allele in this animal successfully went germline. We amplified the entire 1500bp sequence (i.e., included in the F3 to R3 amplicon) of the *Creb5^{lox}* allele from genomic DNA isolated from the F1 generation derived from founder mouse 3; and sequenced this entire fragment. This 1500bp sequence includes both LoxP sites that flank *Creb5* exon 9 and is displayed in Supplementary Table 8.

Some functional work establishing the role of *creb5* in the *wnt5/yap/taz* loop, otherwise the text is merely speculative.

Thanks for pointing this out. We now state:

" While our work indicates that forced expression of *iCreb5-HA^(Prx1-Cre)* in the perichondrium represses that of *Wnt5a* and *Crabp1* in this tissue, it is not clear whether the repression of *Wnt5a/Crabp1* expression is either a direct effect of *Creb5* or an indirect effect of one or more of the down-stream genes induced by *Creb5* in the articular perichondrium (diagrammed in Fig. 10d)."

Reviewer #4 (Remarks to the Author):

The manuscript entitled 'Creb4 is necessary for joint formation and directs the genesis of articular cartilage' by Zhang et al. describes the functions of the transcription factor Creb5 in joint formation during embryonic development using a global knock-out allele and a conditional transgene allowing for tissue-specific Creb5 expression. In addition, the authors provide data on the postnatal function of Creb5 using the TAM-inducible Agg-CreERT2 line. Overall this is an interesting manuscript suggesting that Creb5 plays a role in the formation of the joint interzone and is postnatally required for the maintenance of the superficial layer of the articular chondrocytes. In this manuscript, the authors have also incorporated most of the previous concerns raised by the reviewers. Yet, there are still a few major and minor concerns regarding the interpretation of the results and the mechanism proposed. Furthermore, the figures need some more work (see comments regarding the figures below).

Comments:

- A major concern is that the embryonic skeletal phenotypes described here, in the *Creb5*^{Δ9/Δ9} mice, could be in part be due to defects in muscles. This is a total knock-out and Creb5 has also been reported in the literature to be expressed in the musculature and from the images shown here for Creb5 transcript and protein, it appears that there is some staining in the soft tissue as well. Furthermore, it is known from the literature that spontaneous muscle contractions are required for proper joint formation during embryonic development.

Thank you for raising this interesting possibility. However, the loss of nearly all synovial joints in mice lacking Creb5 function is not consistent with a general loss of skeletal muscle activity in such animals. While Creb5 expression is most intense in the forming interzone of the synovial joint, we acknowledge that this transcription factor is also expressed to some extent in interstitial connective tissues (including tendons and ligaments). However, prior work has indicated that the absence of skeletal muscle activity only affects a limited number of synovial joints and most notably does **not** affect formation of the knee joint (see Muscle Contraction Is Necessary to Maintain Joint Progenitor Cell Fate Joy Kahn, David M. Kingsley,⁵ and Elazar Zelzer, Dev Cell., '09). In contrast to the knee joint, which still forms in animals that lack skeletal muscle function, the knee joint completely fails to form in either *Creb5*^{Δ9/Δ9} mice (with germline deletion of Creb5) or in *Prx1Cre-Creb5*^{fllox9/fllox9} mice (with conditional knockout of Creb5 in limb bud mesenchyme). Thus, the loss of the knee joint following loss of Creb5 function cannot be explained by loss of skeletal muscle function; and is most consistent with the necessity for Creb5 expression in the interzone of the developing synovial joints directing the formation of these structures. In addition, while prior work has noted that muscle activity is necessary for the formation of bone eminences, such as the olecranon and deltoid-tuberosity; we have noted that both the olecranon and deltoid-tuberosity form normally in the absence of Creb5 function (i.e., in *Creb5*^{Δ9/Δ9} mice; Supplementary Figure 2). Thus, the phenotype resulting from the loss of Creb5 function (i.e., loss of most synovial joints, including the knee; but maintenance of the olecranon and deltoid-tuberosity) is quite distinct from the phenotype resulting from loss of skeletal muscle activity (i.e., limited loss of synovial joints with normal knee joint formation, and complete loss of the bone eminences including the olecranon and deltoid-tuberosity). Thanks again for raising this question; we have now discussed this point in the revised discussion.

- As for the transgenic study expressing Creb5 under the control of Prx1Cre, it is surprising that the transgene expression appears rather limited to certain areas and not as expected throughout the entire limb mesenchyme. Immuno-stained sections for the HA-tag of the transgene should be provided.

Thanks for this suggestion. We have extended our analysis of the limb bud expression profiles of both Creb5-HA and GFP transgenes to earlier stage (E11.5) *Prx1-Cre-iCreb5-ires-GFP* embryos (Supplementary Figs. 5a and 5b). While both the Creb5-HA and GFP transgenes were expressed throughout the limb bud mesenchyme in E11.5 *Prx1-Cre-iCreb5* embryos (Supplementary Figs. 5a and 5b), transgene-expressing cells were excluded from the cartilage elements and were restricted to the perichondrium in the limb buds of E14.5 *Prx1-Cre-iCreb5* embryos (Figs. 5a & 5b).

- Based on the whole-mount skeletal preparations of the transgenic animals provided the skeletal elements seemed to be pretty messed up. Hence, in sectioned material, misinterpretations due to the level of sectioning are possible. Unfortunately, some of the sections from the transgenic limbs may not be oriented properly, and hence, some of the skeletal elements may not be cut in the most optimal way. Most relevant, the plane of the section can have an influence on the marker staining. For e.g. the proximal element (humerus) is not visible in the control and only (the distal) part of it appears to be visible in the Prx1-Cre-iCreb5 sample in Figure 5a. Nevertheless, the authors use this example to claim that Wnt5a is no longer expressed in the perichondrium of the proximal element. Furthermore, the sample shown in 5b, looks quite different - why is there no Wnt5a signal (to the same extent) visible in the perichondrium surrounding the distal elements as in 5a (white arrowhead).

Thanks for pointing this out. We have eliminated old Figure 5a, and added New Figures 5a and 5b, which more clearly depicts the mutual exclusion of transgenic Creb5 and endogenous Wnt5a.

Additional perichondrial markers should be used (Bandyopadhyay et al. 2008) to show that the effect is specific to Wnt5a (as claimed by the authors).

Thanks for this suggestion. We have added a new RNA-Scope analysis for Crabp1 (in New Figure 5b) which, like Wnt5a, is expressed in the perichondrium in a reciprocal fashion to Creb5. Thus, the expression domain of Creb5 versus that of Wnt5a/ Crabp1 define two regions in the perichondrium, which give rise to distinct cell fates (i.e., articular cartilage and cortical bone, respectively). We have found that transgenic expression of iCreb5 in the perichondrium (in *Prx1-Cre-iCreb5* embryos) represses perichondrial expression of both Crabp1 and Wnt5a.

- On some of the images of Prx1-Cre-iCreb5 samples, it appears as if there are more cartilage elements than anticipated (see for example Col2 staining in Suppl. Fig. 4). This is also apparent in the whole-mount skeletal preparation shown in Figure 6a. Therefore, it would be helpful if the authors could indicate which skeletal elements are seen on the individual images and to which joint the arrowhead is pointing.

We have labeled the Femur and Tibia in images of the RNA-Scope analysis of E14.5 *Prx1-Cre-iCreb5* and control embryos (New Figure 5b). Because of the gross distortion of the long bones, polydactyly, and excessive production of ankle bones, we were not able to definitively label the

identity of the bones in the whole-mount skeletal preparation shown in Figure 6a, which is from Po mice.

As chondrogenesis appears to be severely altered and delayed in the *Prx1-Cre-iCreb5* limbs, it is very difficult to make statements regarding the intensity of signals such as *Gdf5* and *Wnt4* compared to a control that is more developmentally advanced. Hence, the authors would need to provide complementary data to show that the levels of *Gdf5* and *Wnt4* are elevated in the *Prx1-Cre-iCreb5* limbs.

We have added a new RNA-Scope analysis (in New Figure 5b) for the synovial joint interzone markers, *Gdf5* and *Wnt11*, in E14.5 *Prx1-Cre-iCreb5* embryos and control littermates. Notably, the *iCreb5-HA^(Prx1-Cre)*-expressing perichondrium displayed elevated and ectopic expression of the joint interzone markers *Gdf5*, *Wnt11* and *Wnt4* (Fig. 5b, yellow arrowheads; Supplementary Fig. 6, yellow arrowheads), which usually display nested patterns of expression in the articular perichondrium (Fig. 5b, yellow arrows; Supplementary Fig. 6, yellow arrows).

- How do the authors explain the polydactyly phenotype (which to a certain extent is also visible in the forelimb and not just in the hindlimb)? As pointed out in other places, it appears that additional skeletal elements are also visible in the more proximal region (Figure 6a, Col2 staining in Suppl. Fig. 4). This should be documented better and may argue in support of the idea that *Creb5* can induce an interzone-like structure leading to the segmentation of skeletal elements. For this, it would probably be necessary to look in earlier limbs (such as E11.5-E13.5) and also document the expression of the transgene using the HA-tag better.

Thank you for asking us to explain the polydactyly phenotype observed in *Prx1-Cre-iCreb5* animals. In addition to inducing hindlimb polydactyly, we noted that both forelimb and hindlimb buds were larger in E11.5 *Prx1-Cre-iCreb5* embryos versus their control littermates (Supplementary Fig. 5a). To understand what might cause these phenotypes, we examined the expression of *Msx1* in the hindlimb buds of E11.5 *Prx1-Cre-iCreb5* embryos and control littermates. Prior work has established that BMP signaling promotes the expression of *Msx1* in the limb bud mesenchyme, can drive apoptosis of limb bud mesenchymal cells, and also restricts the expression of *Fgf4/8* in the apical ectodermal ridge (AER). Indeed, manipulations that decrease the level of BMP signaling in the limb bud both promote an expansion of *Fgf4/8* (and the AER) into a larger region of the limb bud ectoderm and induce polydactyly. Interestingly, we observed that misexpression of *iCreb5-HA^(Prx1-Cre)* in E11.5 embryos both expanded the expression of *Gdf5* to a broader domain of the limb bud; and simultaneously decreased the expression of *Msx1* in the limb bud mesenchyme (Supplementary Fig. 5c). As *Gdf5* has been found to block signaling by *BMPR1a*, we speculate that both the increased size of limb buds in E11.5 *Prx1-Cre-iCreb5* embryos, repression of *Msx1*-expression, and hindlimb polydactyly may all reflect an attenuation of BMP-signaling in the limb bud mesenchyme due to ectopic expression of *Gdf5*.

- Limb bud micromass cultures could be performed. Here, the expectation would be that *Prx1-Cre-iCreb5* limbs should give rise to smaller nodules. In this culture, the localization of the HA-tagged transgene should also be analyzed. Based on the hypothesis of the authors it should accumulate in cells surrounding the cartilage nodules.

Thanks for this interesting recommendation. However, we have focused our current efforts on obtaining information regarding the spatial localization of gene expression in *Prx1-Cre-iCreb5* limbs in vivo; rather than extending our analysis to in vitro models that are derived from these

tissues. Future work (which is outside the scope of the present study) will explore whether misexpression of Creb5 in limb bud micromass cultures alters either cellular fate or cell localization.

- The data on the Sox9-IRES-Cre are convincing. Interestingly, bone protrusions such as the deltoid tuberosity and the olecranon appear to be missing or are reduced in these mutants. Whether the patella is also affected cannot be judged from the image provided in Figure 7c.

We have added images of the Alcian Blue and Alizarin Red stained whole mount appendicular skeleton isolated from a *Po Sox9-IRES-Cre-iCreb5* mouse (and a control littermate). The deltoid tuberosity, olecranon, and patella are all present, but diminished in size (Supplementary Figure 7).

- The differences in Fat4 and Wif1 expression patterns in the joint interzone of the Creb5 Δ 9/ Δ 9 mutants do not really support the conclusion by the authors that Creb5 promotes the expression of these two genes. As the joint does not form normally genes expressed in that region can be altered simply as a consequence. The same is true for the in vitro experiments using shRNA for Creb5. This may also lead to an alteration of the cell type. Promotor studies would be needed to show a direct interaction. How do the authors explain the differences in the recovery of the deregulated genes between conditions A and AB.

We have found that the expression of both Fat4 and Wif1 were decreased following shRNA-mediated knockdown of Creb5 in bovine articular chondrocytes. Consistent with this observation, we also observed that the expression of both Fat4 and Wif1 were decreased in the forming synovial joint interzones following loss of Creb5 function. We fully agree that these observations suggest that Creb5 is either necessary to drive the expression of Fat4 and Wif1 in joint interzone cells or is necessary to maintain the interzone cell fate. We now state: " Taken together, our findings indicate that Creb5 is necessary to drive the expression of several signaling molecules in the joint interzone, including Gdf5, Wnt9a, Wnt11, Wif1 and Fat4; and/or maintains their expression by establishing the interzone cell fate. " Future work (which is outside the scope of the present study) will be necessary to determine whether Creb5 binds directly to the regulatory elements that drive expression of either Fat4 or Wif1 in the developing synovial joint interzone.

We think that all genes whose expression was decreased in bovine articular chondrocytes following knockdown of Creb5 are either directly or indirectly regulated by this transcription factor. The fact that some genes display a greater dependency upon Creb5 in either the absence or presence of TGF-beta signaling (A versus AB conditions, respectively), may reflect differential dependency of Creb5 target genes on Smad2/3 function. Prior work from our lab has indicated that Creb5 has the potential to directly bind to Smad2/3, but future work (which is outside the scope of the present study) is necessary to evaluate whether Creb5 and Smad2/3 bind to the regulatory regions that drive the expression of all or only a subset of Creb5 transcriptional targets.

Minor comments:

- Please mention also briefly in the result part, when introducing the mutant allele, that the deletion of exon 9 leads to an in-frame deletion in the CREB5 protein.

Thanks for this recommendation. We now state: " To determine the role of Creb5 in joint formation, we employed CRISPR/Cas9 to produce a mouse line containing an in frame deletion of exon 9 in one allele of *Creb5* (*Creb5*^{Δ9/+} mice; Supplementary Figs. 1a-1d)."

- On page 7, the authors describe that the CREB5Δ9 protein can still be detected in the radiohumeral joint and so can phospho-Smad2. Is this observed in all joints or specific for this particular joint?

We only evaluated Creb5 and phospho-Smad2/3 immunostaining in the elbows of *Creb5*^{Δ9/Δ9} mice (and a control littermate), as this was one of the fused joints in *Creb5*^{Δ9/Δ9} mice in which we still detected residual Prg4 expression (within the fused radiohumeral element). Immunocytochemistry for Creb5 revealed that Creb5(Δ9) protein was expressed in the perichondrium surrounding the radiohumeral fusion in *Creb5*^{Δ9/Δ9} mice (Fig. 3c), suggesting that Creb5 DNA binding is critical for both separation of the radius and humerus and for appropriate localization of Prg4⁺ cells in the elbow. Consistent with the requirement for TGF-β signaling to maintain Prg4 expression in articular cartilage, we noted that carboxy-terminal phosphorylated-Smad2 (which is a proxy for active TGF-β signaling) was also detectable in perichondrium of either control mice or *Creb5*^{Δ9/Δ9} mice (Fig. 3c).

In the knee joint of *Creb5*Δ9/Δ9 specimens (shown in Figure 2d-f), this does not seem to be the case for CREB5Δ9.

We evaluated Creb5 RNA expression in the knees (in Figure 2d-f), which document that Creb5Δ9 RNA expression is significantly depressed in the fused knee joints of *Creb5*^{Δ9/Δ9} mice relative to Creb5-WT RNA expression in the knee joints of WT littermates. In both the fused femorotibial element and the fused radiohumeral element in *Creb5*^{Δ9/Δ9} mice, the expression of both Creb5Δ9 RNA (as analyzed in Figures 2d-f) and Creb5Δ9 protein (as analyzed in Figure 3c) is significantly diminished relative to that of Creb5-WT RNA and Creb5-WT protein in WT littermates.

- Page 8: The concluding sentence 'Thus, not only is Creb5 activity necessary to promote the expression of joint interzone markers, but it is also critical to exclude the expression of Wnt5a from the perichondrium adjacent to the forming joint' is a bit of an overstatement, as it has not been shown that Creb5 activity promotes the expression of any joint marker except for Prg4 which is a rather late marker and not really an interzone marker. Given that the architecture of the joint is lost the abnormal expression of Wnt5a could just be a consequence thereof. Yet, the authors make it sound as if Creb5 activity is directly involved in the regulation of interzone marker genes and of Wnt5a.

We have added a new RNA-Scope analysis (in New Figure 5b) for the synovial joint interzone markers, Gdf5 and Wnt11, in E14.5 Prx1-Cre-iCreb5 embryos and control littermates. Notably, the iCreb5-HA^(Prx1-Cre)-expressing perichondrium displayed elevated and ectopic expression of the joint interzone markers Gdf5, Wnt11 and Wnt4 (Fig. 5b, yellow arrowheads; Supplementary Fig. 6, yellow arrowheads), which usually display nested patterns of expression in the articular perichondrium (Fig. 5b, yellow arrows; Supplementary Fig. 6, yellow arrows). We certainly agree

that it is not clear from our analysis whether Creb5-dependent alterations in gene expression is due to a direct binding of Creb5 to these target genes or due to the induction of intermediate regulators of these genes. We now state: "Taken together, our findings indicate that Creb5 is necessary to drive the expression of several signaling molecules in the joint interzone, including Gdf5, Wnt9a, Wnt11, Wif1 and Fat4; and/or maintains their expression by establishing the interzone cell fate."..." While our work indicates that forced expression of iCreb5-HA^(Prx1-Cre) in the perichondrium represses that of Wnt5a and Crabbp1 in this tissue, it is not clear whether the repression of Wnt5a/Crabbp1 expression is either a direct effect of Creb5 or an indirect effect of one or more of the down-stream genes induced by Creb5 in the articular perichondrium (diagrammed in Fig. 10d). "

- Please indicate how many independent samples have been analyzed to confirm the staining results.

We have added this information to the Figure legends.

- Why were deep zone chondrocytes used for the PRG4 Chip experiments. Prg4 is normally a marker for superficial chondrocytes and the latter was used for the RNAseq experiments.

We employed deep zone bovine chondrocyte (in Supplementary Figure 1) to compare whether lentivirus-encoded Creb5^{WT} protein versus lentivirus-encoded Creb5^{Δ9} protein can either induce the expression of Prg4 or directly bind to the Prg4 regulatory sequences. We employed deep zone chondrocytes, as these cells display relatively low levels of expression of both endogenous Creb5 and Prg4, as opposed to superficial zone chondrocytes, which express relatively high levels of both these genes. Thus, by employing deep zone chondrocytes for both RT-qPCR and ChIP assays, we were able to specifically assay the activities of lentivirus-encoded Creb5^{WT} versus lentivirus-encoded Creb5^{Δ9} without the confounding expression of high levels of endogenous Creb5 and Prg4.

- Page 9: the statement 'Consistent with the requirement for Wnt5a to induce expression of the osteogenic transcription factor Runx2 in the perichondrium (35)....' is incorrect. The paper by Yang et al. 2003 does not show that Wnt5a is required for Runx2 expression. The paper only shows that chondrocyte maturation is delayed in Wnt5a mutants and as a consequence Runx2 expression and osteoblast differentiation are delayed. Accordingly, chondrocyte differentiation may also be delayed in the Prx1-Cre-iCreb5 limbs. This should be analyzed using markers such as Col10a1 and Ihh. Especially the latter has been shown to couple chondrogenesis to osteoblast differentiation in the surrounding perichondrium/periosteum.

Yang and colleagues documented that germline loss of Wnt5a function led to decreased expression of Runx2 in both the perichondrium and in prehypertrophic cells in the cartilage elements of the developing long bones; and that loss of Wnt5a is associated with loss of chondrocyte hypertrophy (reference 36). Thus, this prior work indicated that maintained perichondrial expression of Wnt5a in the developing zeugopod cartilage elements requires functional Wnt5a ligand. We confirmed this result, by examining expression of Wnt5a in embryos containing a deletion of exon 2 in both alleles of *Wnt5a* (*Wnt5a*^{Δexon2/Δexon2}) and employing an RNAscope probe outside of the deleted exon. Notably, perichondrial expression of Wnt5a is extinguished, and expression of its downstream target gene *Ctgf* is significantly reduced, in the zeugopod cartilage elements of E14.5 *Wnt5a*^{Δexon2/Δexon2} embryos (Fig. 6d). In contrast, expression

of *Wnt5a* is still maintained in the autopod region of E14.5 *Wnt5a^{Δexon2/Δexon2}* embryos (Fig. 6d). These findings indicate that *Wnt5a* function is required to maintain its own expression specifically in the perichondrium. Interestingly, the cartilage elements in *Prx1-Cre-iCreb5* embryos which failed to elongate similarly lacked perichondrial expression of *Wnt5a*; and displayed reduced *Ctgf* expression in the adjacent cartilage tissue (Fig. 6e), suggesting that ectopic perichondrial expression of *iCreb5-HA^(Prx1-Cre)* has somehow disrupted a *Wnt5a*-positive feedback-loop. Thank you for asking us to evaluate *Col10a1* expression in *Prx1-Cre-iCreb5* embryos. Consistent with the role for *Wnt5a* to support both chondrocyte hypertrophy and longitudinal bone growth, we found that both the stylopod and zeugopod cartilage elements in E14.5 *Prx1-Cre-iCreb5* limb buds lacked the expression of *Col10a1* (Fig. 5b).

- Figures should be organized in the way they are mentioned in the main text. This is not the case for Figures 6d and 5d, which are discussed after Figure 7. Figure 5d even after the mentioning of Figure 6d.

Thanks for pointing this out. We have reorganized the figures such that they are aligned with the text (as much as possible).

The images shown in Figure 6d are presumably from a forelimb sample (as stated in the figure legend). Yet, this sample shows clear polydactyly and is, therefore, more likely derived from a hindlimb. Furthermore, the elements shown here appear more likely to be carpal or tarsal elements, which would also be surrounded by *Prg4* and *Wif1* positive cells. What the *Col2a1/Matn1* staining would look like on carpal or tarsal elements would need to be analyzed.

The FISH images, now displayed in Figure 7a, are indeed from the forelimb (which also sometimes displays polydactyly in *Prx1-Cre-iCreb5* embryos). While it is difficult to assign the identity of all the cartilage elements in the limbs of *Prx1-Cre-iCreb5* embryos, we identified the proximal-most limb element as the humerus in these images-and have labeled it thus. We display the expression of the various genes as assayed by FISH in Figure 7a in either the humeroulnar joint in WT-littermates or in the entire limb in *Prx1-Cre-iCreb5* embryos. Please note that all these images were taken at the same magnification.

- Regarding the *Wnt5a^{Δex2}* allele, the authors should provide RT-PCR data showing that this is not a transcript null.

Wnt5a^{Δexon2/+} mice were generated by crossing a male *Wnt5a^{flox-exon2/flox-exon2}* mouse ⁷⁷ (purchased from The Jackson Laboratory, Stock No: 026626) with a female *EIIa-cre* mouse (purchased from The Jackson Laboratory, Stock No: 003724). The germline deletion of the floxed exon 2 of the *Wnt5a* allele was confirmed by PCR in *EIIa-cre* negative offspring using the following primers: F: GGTGAGGGACTGGAAGTT and R: TGCTTCAGACACTGTGGC. The lengths of the PCR products for the wildtype *Wnt5a* allele and the *Wnt5a^{Δexon2}* allele (after deletion of exon 2) are 832bp and 330 bp, respectively. *Wnt5a^{Δexon2/+}* males were crossed with *Wnt5a^{Δexon2/+}* females to generate litters containing both *Wnt5a^{Δexon2/Δexon2}* embryos and control embryos (which were either *Wnt5a^{Δexon2/+}* or *Wnt5a^{+/+}*). We determined the sequence (and added this information to our methods) of the 330bp PCR product that identifies the *Wnt5a^{Δexon2}* allele as (primer to primer): GGTGAGGGACTGGAAGTTGCAGGAAAGAATTCGGCGCGCCATAACGGGGGTATAGGATACAT TATACGAAGTTATTTAATTAAGGCTACCATGGAGAAGTTACTATTCCGAAGTTCCTATTCTCTA GAAAGTATAGGAACCTCAAGCTGATCCTAAGCTTGCGCGGCCGCCCGCCCGCACCCCCTCACCAGG TAAAATAAGAGAGAGTAGCCGTATTATTTTCTGTGCTTTGAGGTTATTAAGCCACAGTGTGTC

CCCCAAGGTAAAATAAAAATGAGTTTCCCTTTTATTTTTCTGTGCGTTTGAGGTTATTAAGCCAC
AGTGTCTGAAGCA.

- The supplementary tables 1 and 2 are currently provided as excel files. This excel files need to be reorganized into proper columns and the different treatment conditions need to be indicated. Currently, it is not possible to see which excel corresponds to which treatment.

Thanks for the recommendation. The titles of the tables have been added to these files which indicates the treatment regimen; and we have reorganized the data such that it appears in proper columns in "Excel"...sorry about this oversight. Previously, the data was formatted for viewing in "Numbers".

- Images of E14.5 knee joints should be provided for the conditional Prx1-Cre mediated knock-out of Creb5, as these allow better comparison with the straight knock-out.
- RNAscope analysis using a probe directed against exon9 of Creb5 should be performed to show deletion of Creb5 in the postnatal sample. Images showing postnatal expression of Creb5 in the joints should also be provided.

Thank you for this recommendation. Due to the low level expression of Creb5 in the articular cartilage of P14 and older mice, we have been unable to directly detect full length Creb5 by RNA-Scope in mice that are P14 or older. Thus, we have added new results which employ RT-qPCR analysis to assay Cre-mediated deletion of *Creb5* exon9 in postnatal *Aggrecan1^{tm(IRES-CreERT2);Creb5^{flox9/flox9}}* mice. To study the role of Creb5 in postnatal synovial joints we generated mice containing Creb5 alleles in which exon 9 (which encodes the bZIP DNA binding domain) is flanked by loxP sites (*Creb5^{flox9/flox9}* mice; Supplementary Figs. 9a & 9b). Like *Creb5^{Δ9/Δ9}* animals, *Prx1-Cre; Creb5^{flox9/flox9}* mice also displayed an absence of a knee joint (Supplementary Fig. 9c), indicating that the floxed alleles in *Creb5^{flox9/flox9}* mice can be deleted by Cre-mediated recombination. We mated *Creb5^{flox9/flox9}* mice with *Aggrecan1^{tm(IRES-CreERT2);Creb5^{flox9/flox9}}* mice to generate litters containing both *Aggrecan1^{tm(IRES-CreERT2); Creb5^{flox9/flox9}}* mice or *Creb5^{flox9/flox9}* control mice (which lack the CreERT2 driver allele). Tamoxifen was repeatedly administered to these litters at postnatal days 1-11, to specifically delete exon 9 of *Creb5^{flox9/flox9}* in postnatal stage articular chondrocytes. Following sacrifice at P14, RT-qPCR analysis of RNA isolated from the femoral heads of *Aggrecan1^{tm(IRES-CreERT2);Creb5^{flox9/flox9}}* mice indicated that exon9 of *Creb5^{flox9/flox9}* was indeed deleted in femoral head articular cartilage (Supplementary Fig. 9d). RNAScope analysis of gene expression in the knees of these animals revealed that expression of both *Prg4* and *Wif1* was markedly decreased in P14 *Aggrecan1^{tm(IRES-CreERT2);Creb5^{flox9/flox9}}* mice in comparison to their control *Creb5^{flox9/flox9}* littermates (Fig. 9a). In striking contrast to the loss of both *Prg4* and *Wif1* expression following deletion of *Creb5* in the forming joint, the expression of *Ctgf* was increased in both periarticular chondrocytes and in the metaphyseal perichondrium (Fig. 9a). Due to the low-level expression of Creb5 in the articular cartilage of P14 and older mice, we have been unable to directly detect full length Creb5 by RNA-Scope in mice that are P14 or older. However, we know that Creb5 is indeed expressed in the articular cartilage in P14 animals because we can detect Creb5 transcripts in P14 femoral head articular cartilage via RT-qPCR. In addition, we have recently generated a *Creb5*-P2A-tdTomato mouse, which contains tdTomato that has been knocked into the 3'UTR of *Creb5*. Consistent with our prior work indicating that *Creb5* is expressed in the superficial zone of the articular cartilage in adult human femoral head articular cartilage (reference 18), we have observed the expression of tdTomato in the superficial-most cells of the articular cartilage in P14 *Creb5*-P2A-tdTomato mice.

- The layer of flat chondrocytes appears to be still present in the articular cartilage of the upper skeletal element (distal end of the femur?) in the AggCreERT2;Creb5fl/fl samples. Please indicate in the images in Figures 10a and b what is the femur and what tibia. The articular cartilage in the P42 samples of controls and AggCreERT2;Creb5fl/fl samples does not appear to be very different.

We have indicated the surface of the femur, tibia, and meniscus in Figures 10a and 10b. We have found that the number of flat cells located in the superficial zone are substantially reduced on the articular surface of both the femur and the tibia of tamoxifen treated AggCreERT2;Creb5fl/fl mice (Figures 10a and 10b); and also reduced in number on the surface of the anterior meniscus in these mice (Figure 9c). We have also added Col2a1 RNA-Scope which better documents that the Col2a1-expressing flat cells in the superficial region of the articular cartilage are decreased in number by approximately 75% in both the femoral and tibial surface of the knee joint in such tamoxifen treated AggCreERT2;Creb5fl/fl mice (which are now quantified in Figure 10c). In addition, we have established that postnatal loss of Creb5 function in the articular cartilage leads to a loss of both Prg4 and Wif1, which are diagnostic of the superficial zone of the articular cartilage in both the femur and tibia in the knee joint; and added additional evidence that postnatal loss of Creb5 in the articular cartilage leads to both a premature loss of Col10a1 expressing cells in the secondary ossification center (Figure 9b) and to premature ossification of the anterior meniscus (Figure 9c).

- According to the publication by Gamer et al. 2016 the anterior (4 weeks) and posterior menisci (8 weeks) mineralize at different timepoints in mice. Hence, it would be good if the authors make sure that they compared the same menisci (currently there is no information provided in the figure legend on which part of the meniscus is shown). Images of sections in which both menisci are visible may be more useful here (or micro CT images of the whole knee).

We are showing images of the anterior meniscus at both P30 (which displays evidence of premature chondrocyte hypertrophy) and at P42 (which displays premature ossification) following deletion of Creb5 in AggCreERT2;Creb5fl/fl mice. We have observed evidence of premature endochondral ossification of the anterior meniscus in 3/3 AggCreERT2;Creb5fl/fl mice that were treated with tamoxifen (Figure 9c).

- Global misexpression of Wnt5a in chicken limbs (Hartmann and Tabin, 2000) or throughout the Col2a1-expressing domain (including chondrocytes at the joints; Yang et al. 2003) did not lead to joint alterations as would be expected based on the model of the mechanism proposed here. Can the authors explain this?

Our findings indicate that misexpression of Creb5 specifically in limb bud mesenchymal cells leads to loss of Wnt5a expression in the perichondrium; and that loss of Creb5 leads to an expansion of Wnt5a expression in the perichondrium into the joint forming area. Together these findings indicate that Creb5 either directly or indirectly represses expression of Wnt5a in the articular perichondrium. However, our findings do not shed light on what signals regulate the region-specific expression of Creb5 in the articular perichondrium. Because we have no evidence to indicate that ectopic Wnt5a can inhibit Creb5 expression, we do not suggest that such is the case in the model displayed in Figure 10d.

- Regarding the proposed hypothesis that Creb5 in the articular perichondrium may influence the timing of ossification, the authors should provide evidence that the onset of SOC formation

is altered in conditional Prx1Cre-Creb5fl/fl specimens. Currently, this is highly speculative and based on an embryonic phenotype that can be interpreted in alternative ways.

We have now included RNA-Scope analysis of Col10a1 expression in the knees of tamoxifen treated P30 Aggrecan-CreERT2/Creb5floxed/floxed mice (Figure 9b). We observed that loss of Creb5 results in a decreased number of Col10a1 expressing cells on the side of the secondary ossification center that is nearest the articular surface. This observation is consistent with the notion that Creb5 induces the expression of a signaling molecule that promotes the maintenance of Col10a1 expressing cells and blocks either their premature loss and/or maturation to join the secondary ossification center. Consistent with the notion that Creb5 indirectly blocks premature endochondral ossification, we have also observed evidence of premature endochondral ossification of the anterior meniscus in 3/3 AggCreERT2;Creb5fl/fl mice that were treated with tamoxifen (Figure 9c).

- Some of the images shown in Figure 3c are identical to images shown in Figure 2e from the previous publication by Zhang et al. 2021 and hence, may not correspond to littermate controls.

We previously published a portion of the data in Figure 3c, that specifically assayed gene expression in only the control littermate from this figure.

- In their previous paper, the authors already provided experimental data using the iCreb5HA construct that Creb5 can in combination with TGFbeta2 and TGFalpha upregulate Prg4 expression. Yet, relative expression levels in the previous experiments were increased in the presence of TGFbeta2 alone only about 18-fold, and here around 70-fold, while cotreatment with TGFbeta2 and TGFalpha previously led to an increase in the levels by about 44-fold (here almost 80-fold). In both experiments, only two independent samples have been analyzed. The huge discrepancy between these independent experiments is slightly alarming.

Our experiments involve lentivirus infection of primary cultures of deep zone articular chondrocytes isolated from the knees of genetically diverse cows. While we try to obtain similar types of tissues in these various preparations, there is likely to be some variability in these materials, as these are all primary cells from a genetically diverse source. Indeed, these differing chondrocyte preps frequently express differing amounts of endogenous Creb5, as they are from differing dissections. Thus, we attribute the different levels of Prg4 in response to exogenous viral-encoded Creb5 (cultured in the presence of exogenous TGF-alpha and TGF-beta) to reflect the heterogeneity of these primary cells. Most importantly, the data presented in Supplementary Figure 1g documents that in contrast to Creb5-WT which can drive robust Prg4 expression in deep zone articular chondrocytes cultured in TGF-alpha and TGF-beta, mutant Creb5-(Δexon9) is unable to activate the expression of Prg4 in deep zone articular chondrocytes when cultured under similar conditions.

Remarks regarding the figures:

- It would be helpful if the authors could reorganize the figures in a way that the limb structures in all images are always orientated in the same way throughout all figures (like proximal to the left and distal to the right or vice versa).

Thanks for this recommendation. We have reorganized the figures such that the limb structures are oriented proximal to the left and distal to the right.

- Also labeling should be more consistent: same font size, size bars either in all images or always in the right-side panels (if all images of that panel are taken at the same magnification), embryonic stage, and anatomical region should be consistently indicated within a figure.

Thanks for this recommendation. We have reorganized the figures such that embryonic stage, and anatomical region are consistently indicated within a figure, and have moved the scale bars to the right-side panels.

- Additional labels would be helpful within the figures pointing out the specific structures mentioned in the text when describing the phenotype as for non-experts some of the described changes may be difficult to spot.

Thanks for this recommendation, we have added additional notations to the figures so that the structures are more easily identified.

- Please indicate in all figure legends which skeletal elements or limbs are shown. This is for example not clear from Figure 5a. Furthermore, regarding Fig. 5a, why is the staining for the control in the left panel (*Gdf5*) different from the *Prx1-Cre-iCreb5* sample (*iCreb5*). What is the genotype of the control littermates in this case? The overall quality of Fig. 5a is poor as the skeletal elements are very difficult to see. The high perichondrial expression of the transgene is also not visible in that image. Maybe the authors can provide immuno-stained images using the HA tag to visualize specifically the expression of the transgene in these samples.

We have reorganized the figures such that embryonic stage, and anatomical region are consistently indicated within a figure. We have eliminated old Figure 5a, and added New Figures 5a and 5b, which more clearly depicts the mutual exclusion of transgenic *Creb5* and endogenous *Wnt5a* at E14.5. We also show immunostaining of both transgenic *Creb5* and the HA-Tag to be located predominantly in the perichondrium of *P0 Prx1-Cre-iCreb5* mice in New Figure 7c.

- Figure 6c: Here it would be better to pick regions from the skeletal element to the left (which contains a mineralized region) instead of showing regions from the skeletal element which may all be tarsal-like bones and hence would look also undifferentiated in the control.

Thank you for pointing this out. We have reorganized Figure 6c to compare chondrocyte morphology specifically in the femur of the hindlimbs of either a *P0 Prx1-Cre-iCreb5* mouse or a control littermate.

- Suppl. Fig. 2c: *Prg4* staining in the forelimb is shown in a phalangeal element not in carpal elements as suggested by the title. Carpal elements should be labeled so that the reader can appreciate the joint alterations of certain carpal elements. Also, the tarsal elements should be labeled, as currently, it is difficult to actually see which elements are shown in the images.
- High magnification images of the distinct regions indicated by the arrows and arrowheads would be helpful.

Thanks for pointing this out. We now identify both the radius and the radial element in the forelimb autopod; and the tibia and talus in the hindlimb autopod, in both the *Creb5^{Δ9/Δ9}* mice and the control littermate in Supplementary Figure 3c.

Additional comments:

Some of the histological images lack scale bars (e.g. Figure 3)

We have added scale bars for the histological sections in Figure 3.

In some places, it looks like 'PO' is spelled 'Po' (e.g. page 6).

Pzero appears as "Po" in Georgia font.

Spelling mistake on page 13 and in the legend to Suppl. Table 2: TFGalpha instead of TGFalpha

Thanks for catching these spelling errors.

Reviewer #3 (Remarks to the Author):

The authors have, where they feel possible, addressed my previous comments.

I have no further comments.

Reviewer #4 (Remarks to the Author):

The authors have addressed all concerns to full satisfaction. I would like to congratulate them to this nice piece of work.